# Temporal dynamics of the neural representation of hue and luminance polarity

Katherine L. Hermann[1,4,6], Shridhar R. Singh[1,6], Isabelle A. Rosenthal [1,5,6], Dimitrios Pantazis [2] &
Bevil R. Conway [1,3 ✉]

Hue and luminance contrast are basic visual features. Here we use multivariate analyses of magnetoencephalography data to investigate the timing of the neural computations that extract them, and whether they depend on common neural circuits. We show that hue and luminance-contrast polarity can be decoded from MEG data and, with lower accuracy, both features can be decoded across changes in the other feature. These results are consistent with the existence of both common and separable neural mechanisms. The decoding time course is earlier and more temporally precise for luminance polarity than hue, a result that does not depend on task, suggesting that luminance contrast is an updating signal that separates visual events. Meanwhile, cross-temporal generalization is slightly greater for representations of hue compared to luminance polarity, providing a neural correlate of the preeminence of hue in perceptual grouping and memory. Finally, decoding of luminance polarity varies depending on the hues used to obtain training and testing data. The pattern of results is consistent with observations that luminance contrast is mediated by both L-M and S cone sub-cortical mechanisms.

[1] Laboratory of Sensorimotor Research, National Eye Institute, Bethesda, MD 20892, USA. [2] McGovern Institute for Brain Research, Massachusetts Institute of Technology, Cambridge, MA 02139, USA. [3] National Institute of Mental Health, Bethesda, MD 20892, USA. [4] Present address: Department of Psychology, Stanford University, Stanford, CA 94305, USA. [5] Present address: Division of Biology and Biological Engineering, California Institute of Technology, Pasadena, CA 91125, USA. [6] These authors contributed equally: Katherine L. Hermann, Shridhar R. Singh, Isabelle A. Rosenthal. ✉email: bevil@nih.gov

The most basic computations performed by the visual system yield hue and luminance contrast. Hue is that property of color referred to by a color name (e.g., "blue", "green"). Luminance contrast is related to how light or dark a color is, which is, importantly, distinct from absolute luminance. Although hue and luminance contrast are often depicted as independent dimensions[1], the relationship between these features is not well understood[2,3], and the extent to which luminance is carried by both cone-opponent retinal mechanisms (L-M and S) remains unclear[4–7].

On the one hand, the demonstration of independent frequency shifts for color and luminance[8] and the selective impact of adaptation on detection thresholds[9,10] suggest that hue and luminance contrast are processed by separate neural channels, which may be encoded by the separate magnocellular and parvocellular channels in the lateral geniculate nucleus[11]. Separate encoding is also evident in convolutional neural networks trained for object recognition, which show independent filters for hue and luminance contrast in the earliest layer[12–15]. Color and luminance edges are also relatively independent in natural scene statistics[16]. Moreover, hue and luminance contrast are distinguished by their efficiency in visual memory tasks: short-term visual memory is better for color than for luminance contrast when stimuli are matched in cone contrast[17].

On the other hand, the many perceptual interactions of hue and luminance contrast suggest these features are processed together. These interactions are evident in the perception of orientation[18], saturation[19,20], judgments of brightness across colors[21], color categorization[22], color naming[23–25], steady-state visual evoked potential measurements[7], and color decoding with magnetoencephalography[26]. In addition, hue can mask changes in luminance contrast[27]. Consider eight colored spirals: four hues of both light and dark luminance polarity (Fig. 1a). Most people will group them by hue (rows), not by the sign of luminance contrast (columns). Taken together, these data underscore long-standing questions about the extent to which hue and luminance contrast are encoded by the same or separate neural mechanisms and the timing of these operations.

A prominent theory, which reflects the interactions of hue and luminance contrast evident in perception, is that within visual cortex, hue information is processed not in isolation, but together with information about luminance contrast[28]. This theory is supported by the observation that parvocellular neurons in the lateral geniculate nucleus[29] as well as most cells in V1[30–32] respond to both color and luminance contrast. Joint processing of luminance and color would predict that these features are encoded with the same timing. But a small population of V1 cells show striking responses to color stimuli that lack luminance contrast[32–35]. Thus the V1 neurophysiology cannot rule out the possibility that hue and luminance contrast could be encoded by parallel pathways[36], which may contribute differentially to processing in parallel routes through extrastriate areas[37–41]. If hue and luminance contrast are encoded separately, one might expect luminance contrast to be computed earlier than hue because magnocellular neurons have shorter latencies than parvocellular neurons. But because there are relatively fewer magnocellular neurons, their latency advantage may be lost through convergence in visual cortex[42].

Clues to how color and luminance contrast are processed by the brain have been provided by univariate visual evoked potential measurements to equiluminant and achromatic stimuli[43,44]. But it has not been possible to infer from these experiments the underlying neural mechanisms because all subcortical channels respond to nominally equiluminant stimuli[45]. Moreover, such experiments are inconclusive about timing because response latencies depends on stimulus contrast, and

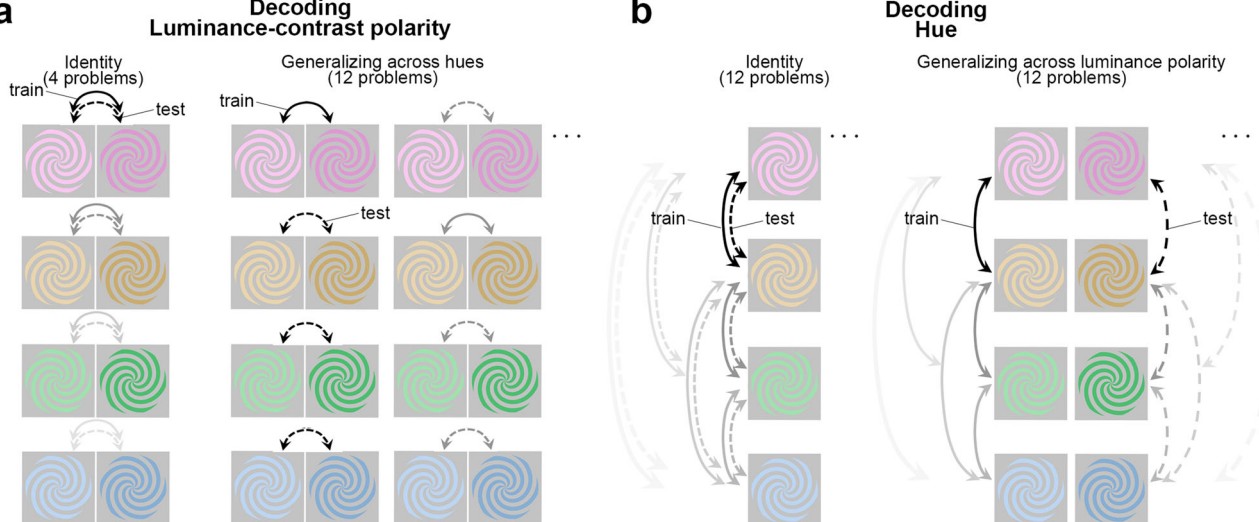

**Fig. 1 Decoding luminance polarity and hue from MEG data: approach. a** Decoding luminance polarity. Participants were shown four hues that varied in luminance polarity (light/dark). Classifiers were trained (solid arrows) and tested (dashed arrows) using stimuli of the same hue (identity problems) or different hue (generalization problems) to determine the extent to which the MEG response to a given color is informative of the luminance polarity carried by the same hue (4 identity problems) or by other hues (12 generalizing-across-hue problems). Each binary classifier was trained to distinguish whether a light or dark stimulus had been presented, given patterns of magnetoencephalography (MEG) sensor activations. For the four identity problems, classifiers were trained and tested on brain responses to the same hue (all four identity problems are illustrated in the graphic). For the generalization problems, classifiers were trained and tested on brain responses to different hues (half of the 12 generalizing-across-hue problems are illustrated). In the graphic, solid lines indicate comparisons used for training and dashed lines indicate comparisons used for testing; line shading distinguishes different problems. **b** Decoding hue. Format as in panel **a**. For stimuli of a given luminance polarity (e.g., dark), a binary classifier was trained to determine which of two hues (e.g., pink or orange) had been presented. The classifiers were then tested, again on held-out trials in which the luminance polarity (e.g., dark) was the same as at train time (12 identity problems) or in which the luminance polarity was opposite (e.g., light), requiring generalization of hue across luminance polarity (12 generalizing-across-luminance-polarity problems).

there is no single method for equating color contrast and luminance contrast[46]. Clues have also been provided by fMRI[47–49], but fMRI does not predict results from behavioral adaptation[10] and cannot uncover the relevant temporal dynamics because it is limited by the relatively sluggish time course of blood flow[50].

In this work, we explore another approach to address the neural mechanisms for color, analyzing the MEGco dataset[26] (MEG color decoding dataset) using multivariate analysis to isolate common and separable representations for hue and luminance contrast. The MEGco dataset was obtained to address two sets of questions. The first set of questions, addressed by Rosenthal et al.[26], concerns the similarity relationships of the patterns of neural activity elicited by different colors. That analysis shows that hue and the sign of luminance contrast (light versus dark) interact in the representation of color, and these interactions are consistent with color-naming patterns: hues labeled with different names for light and dark versions (e.g., yellow/brown) elicit less similar patterns of MEG activity for the light and dark versions, compared to hues that are given the same name across lightness (e.g., green). The second set of questions, addressed presently, concerns the extent to which representations of hue and luminance polarity are separable, and importantly, the temporal dynamics of the representations for hue and luminance polarity. One advantage of approaching these questions using a multivariate analysis is that it uncovers information in a neural representation independent of the magnitude of the response[51], which mitigates confounds that might be introduced by differences in contrast among the stimuli used to elicit the representations. Can hue be decoded independently of luminance polarity, and if so with what timing? Can luminance polarity be decoded independently of hue, and if so with what timing? To the extent that hue contributes to representations of luminance polarity, what is the relative contribution of the two subcortical color channels (L-M and S)? And to what extent do separable representations of hue and luminance polarity generalize across time? Although these questions are related to ones addressed in our previous report, they are distinct and cannot be answered using previously published analyses. The results here show that hue and luminance polarity can be decoded independently from MEG activity, with hue reaching peak decoding about 20 ms after luminance polarity and having slightly greater cross-temporal decodability. We also report here source localization of the MEG data to functionally defined regions identified with functional magnetic resonance imaging (fMRI) in the same participants in whom the MEG data were obtained.

## Results

The experiments were designed to enable a decoding analysis to answer two overarching questions. Given patterns of MEG activity, is it possible to independently decode the hue and luminance polarity (light or dark) of the stimulus that elicited the activity? And what is the time course—that is, how long does it take the brain to extract these features? We addressed these questions by analyzing MEG responses obtained from 18 participants while they were shown spirals that could appear in one of eight colors that varied in hue and luminance polarity (either a luminance increment or decrement) (Fig. 1).

The stimuli were suprathreshold and roughly matched in absolute color contrast and luminance contrast, in terms of detection-threshold units (contrasts were >30× detection threshold, see "Methods" section). Equating stimuli in units of detection threshold is one method among many for comparing color and luminance responses[52–55]. The challenge in comparing responses to color and luminance stimuli arises in part because response magnitude can depend on stimulus contrast[43], especially for near-threshold stimuli. Multivariate analyses of suprathreshold stimuli ameliorate this problem because they uncover any difference in the pattern of response between a pair of variables, providing insight about the information processed by the brain[51].

The colors were defined by the MB-DKL color space[56,57], which is constructed from the cone-opponent dimensions that correspond to retinal color-encoding mechanisms (Supplementary Fig. 1). The selection of stimuli was made not only to tease apart the representations of hue and luminance polarity but also to assess the relative contributions of the two subcortical channels (L-M and S) to representations of luminance polarity. The stimuli were located in the intermediate directions of the space; thus all stimuli involved modulations of both L-M cone activity and S-cone activity, and all stimuli had the same magnitude of modulation of L-M, and the same magnitude of modulation of S. Different colors were created by pairing different signs of these modulations. The luminance contrast of all the stimuli was fixed (25%) but varied in polarity (light or dark) relative to the adapting background. If hue and the polarity of luminance contrast are encoded by separable neural mechanisms, it should be possible to decode hue even if the MEG data used to train the classifiers were elicited by stimuli that differed in luminance polarity from the test stimuli; and it should be possible to decode luminance polarity even if the classifiers were trained using data elicited by stimuli that differed in hue from the test data. In other words, the results should show evidence that hue decoding generalizes across luminance polarity, and luminance-polarity decoding generalizes across hue. The time course should tell us about the relative stage in the visual-processing hierarchy at which hue and luminance polarity representations are encoded and/or the relative amount of recurrent processing required for each computation. Alternatively, if hue and luminance polarity are encoded together, it should be possible to decode specific hue-luminance combinations, but not each dimension invariant to the other dimension. The relative contribution of the two subcortical channels to luminance polarity should be evident using luminance-polarity classifiers trained and tested with data elicited by different hues. Given these various objectives, it was important for the experiment to have enough power, which we ensured by first conducting an extensive pilot experiment to determine the number of trials needed to obtain reliable data (Supplementary Fig. 2).

We used a maximum correlation coefficient classifier (as implemented in the Neural Decoding Toolbox[58], see "Methods" section). We performed within-participant decoding (trained and evaluated classifiers on data in each participant), independently for each timepoint (applied independent classifiers at each time point relative to stimulus onset). All analyses involved pairwise comparisons (Fig. 1) and were cross-validated (Fig. 2), yielding plots that show how representations unfold over time. Participants were told to maintain fixation throughout stimulus presentation and to blink at designated times. Data during eye blinks or breaks in fixation were removed (see "Methods" section). To control fixation and attentional state, participants engaged in a 1-back hue-matching task: every 3–5 trials, the participants were queried with a "?" on the screen to report via button press whether the two preceding stimuli matched in hue. The impact of task was assessed in a control experiment, in which two participants performed half the trials using a 1-back hue matching task and the other half of the trials using a 1-back luminance-contrast matching task. Task had no impact on the main conclusions, as described in the section "Impact of task on decoding." We used a spiral-shaped stimulus to avoid cardinal or radial response biases[59–61].

The eight colors ("conditions"; Fig. 1) were presented in pseudo-random order for 116 ms with 1 s of the gray background

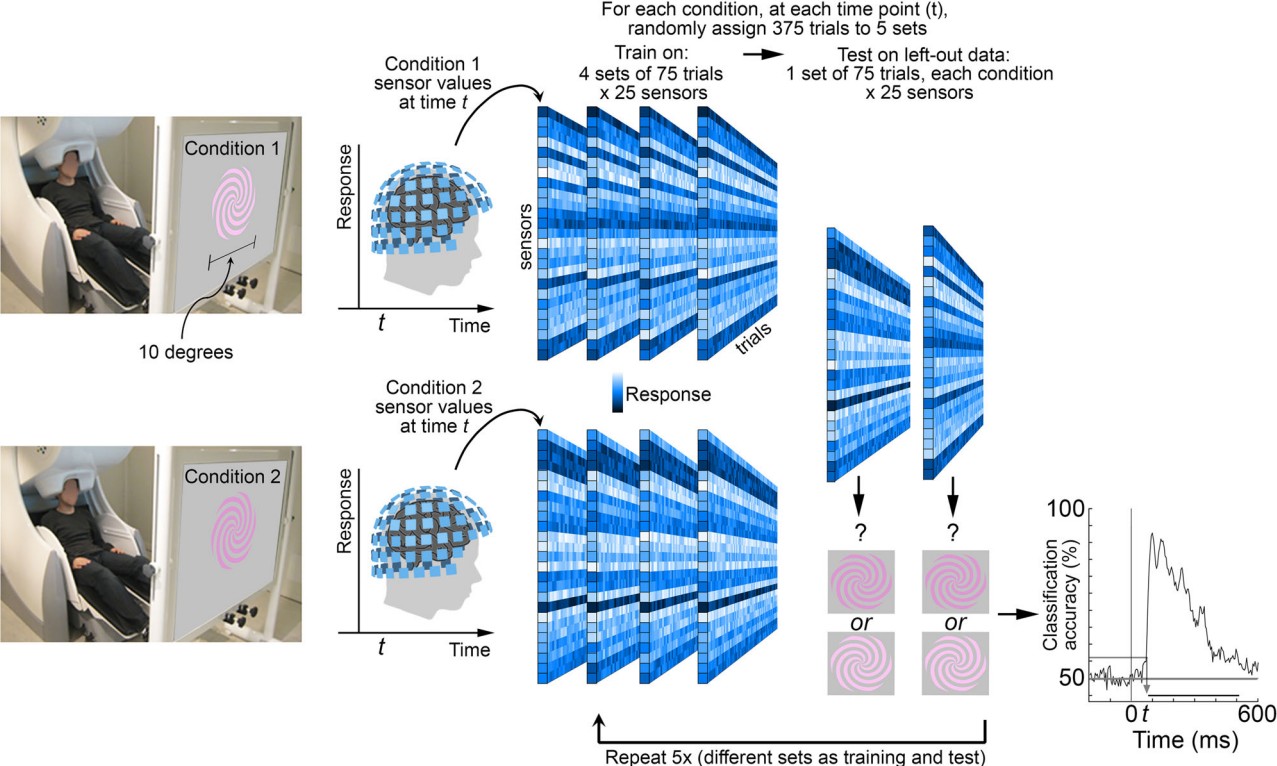

**Fig. 2 Experimental paradigm.** Participants were scanned with magnetoencephalography (MEG) while they looked at colored spirals that were flashed on the screen for 116 ms (with 1 s between flashes). The spirals were one of eight colors: four hues of either a luminance increment or luminance decrement (Supplementary Fig. 1 provides the color specifications for the stimuli). Stimuli were pseudo-randomly interleaved, with 500 presentations of each stimulus over two recording sessions for each of 18 participants. Trials during eye blinks or other artifacts were removed, and the remaining trials were randomly subsampled to yield 375 trials per condition. Sensor data were averaged into 5 ms bins, 200 ms before stimulus onset to 600 ms after stimulus onset. The figure depicts the experimental set up and simulated data to illustrate the analysis pipeline. Classifiers were trained at every time point, and individually for each participant: at each time point ($t$) in the 800-ms time window, the 375 trials were divided into five sets of 75. Four sets were used to train the classifier, and one set was used to test the classifier. The procedure was repeated for the five cross-validation splits; and the entire procedure was repeated 50 times, with different random assignments of the 375 trials into the five sets, yielding a decoding curve showing classification accuracy as a function of time after stimulus onset (bottom right showing actual data for classifying luminance polarity using pink). The horizontal sequence of data points above the $x$-axis shows the time points when decoding was significant for more than four consecutive time points (FDR-corrected; the gray arrow shows the classification accuracy and the time point at which decoding became significant).

between presentations (1 s inter-stimulus interval). We collected responses to a very large number of trials of each condition ($N = 500$), removed trials with artifacts such as eye blinks, and randomly subsampled the remaining trials to obtain 375 trials per condition. Figure 2 illustrates an analysis in which classifiers were trained to decode luminance polarity given patterns of MEG data elicited by bright and dark pink (the decoding curve at the bottom right is real data, the rest of the figure is for illustration purposes). The classifiers were tested on separate data elicited by the same stimuli, bright and dark pink. The results reveal the classification accuracy for luminance polarity carried by a specific hue (pink).

The decoding problem shown in Fig. 2 is referred to as a luminance-polarity identity problem because the hue of the stimuli from trials used to train the classifier is identical to the hue of the stimuli in the test trials. Figure 1a, left, shows schematically the full set of luminance-polarity identity decoding problems: the stimuli associated with the training of classifiers are indicated by solid arrows, and the stimuli associated with the corresponding tests are shown by dashed arrows; shading of the arrows links the training and testing conditions for a given problem. In other problems of luminance-polarity decoding, the hue of the stimuli differed between the trials used to train and the trials used to test the classifier. For example, classifiers were trained using patterns of MEG activity elicited by light and dark pink but tested using

activity elicited by light and dark blue, or light and dark orange, or light and dark green. We refer to these problems as generalizing-across-hues since they uncover the extent to which luminance polarity can be decoded independent of hue (Fig. 1a, right).

In other analyses, we determined the extent to which classifiers could decode hue identity (Fig. 1b, left), and hue generalizing-across-luminance-polarity (Fig. 1b, right). The generalization problems provide a test of invariance: decoding luminance polarity invariant to hue, and decoding hue invariant to luminance polarity. Supplementary Fig. 3 shows the test–retest reliability of the data from the main experiments[62]. All analyses involved binary classifiers (chance is 50%) to facilitate a direct comparison of the results among the different problems.

**Decoding luminance polarity**. Figure 3a shows the average classifier performance across the four luminance-polarity identity problems (solid line) and the 12 generalizing-across-hues problems (dashed line; shading around each trace shows the bootstrapped standard error). The results show that classification accuracy was significantly above chance for luminance polarity for both types of problems. Thus it was possible to decode luminance polarity independent of hue, which supports the hypothesis that the brain has a representation of luminance

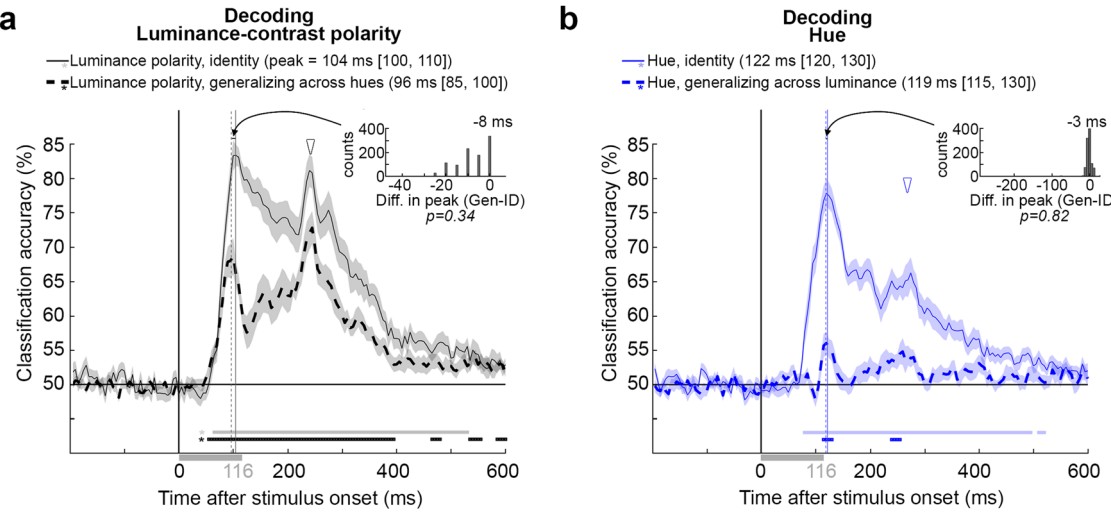

**Fig. 3 Decoding luminance polarity and hue from MEG data: results. a** Classification accuracy as a function of time after stimulus onset for the luminance-polarity problems (center of error shading shows the average of identity problems, solid line; average of generalization problems, dashed line). Traces were generated by averaging 1000 bootstrapped samples across 18 participants. Shading shows the SE of the bootstrap samples. The stimulus duration was 116 ms (gray bar). Inset shows the difference in peak for the 1000 bootstrapped comparisons: identity minus generalization (mean = −8 ms, $p = 0.34$; one-sided test of the proportion of these differences that was less than or equal to zero, not corrected for multiple comparisons). Initial peak of the identity problems, solid vertical gray line, 104.4 ms [100, 110]; initial peak of the generalization problems, dashed gray line, 96.5 ms [85, 100]. Open arrowhead shows the second decoding peak, which corresponds to stimulus cessation. The horizontal sequence of data points above the x-axis, demarcated by asterisks, show time points at which decoding was above chance (determined by a permutation test across subjects and cluster corrected, see "Methods" section); onset of significant decoding for the identity problems was achieved at 65 ms [60, 80], and for the generalization problems at 55 ms [50, 70]. **b** The average performance across 12 sets of identity problems (solid line) and 12 sets of generalization problems (dashed line). The inset shows the difference in peak (for the 1000 bootstrapped comparisons across participants, identity minus generalization, mean = −3 ms, $p = 0.82$, proportion of differences that was less than or equal to zero, not corrected for multiple comparisons). The time to peak was the same for the identity and generalization problems (identity problems: solid vertical line, 121.8 ms [120, 130]; generalization problems: dashed vertical line, 119.1 ms [115, 130]). Onset of significant decoding for the identity problems was achieved at 80 ms [75, 85], and for the generalization problems, at 115 [35, 355]. Other conventions as for panel **b**.

contrast that is independent of the representation of hue. The magnitude of decoding does not provide a measure of the absolute size of an effect, but it is nonetheless a valid measure of relative effect sizes within a given study[51]. Peak decoding accuracy was higher for the identity problems compared to the generalization problems (84% [81, 87] versus (69% [65, 74]; square brackets contain the 95% CI obtained by bootstrapping). This result supports the hypothesis that the brain not only has a representation of luminance polarity that is generalized across hues, but also has a representation of luminance polarity that is combined with the representation of hue. In other words, these results show that the brain has a representation of luminance polarity carried by channels that are hue-selective.

Let us compare the time course of decoding for the identity problems and the generalization problems. The latency at which decoding became significant was not different for the identity problems (65 ms [60, 80]) versus the generalization problems (55 ms [50, 70], based on bootstrapping, $n = 18$, $p = 0.058$). The time to peak was not different for the identity problems compared to the generalization problems (inset Fig. 3a, $p = 0.34$; dashed vertical line, 96.5 ms [85, 100], versus solid vertical line, 104.4 ms [100, 110]; confidence limits computed by 1000 bootstrap draws across participants). Close inspection of the data shows a trend in favor of an earlier onset time and earlier time to peak for the generalization problems (generalization problems showed equal or earlier time to peak than identity problems, $p = 0.002$); this observation implies that the time-to-peak is a reliable measure of how much time the brain takes to generate a representation because it does not vary in a trivial way with changes in the amplitude of decoding (one might have expected time-to-peak to be longer for curves with lower amplitude decoding; the data show, if anything, the opposite). Following peak decoding, the

generalization problems showed a pronounced dip. Finally, both the identity and generalization problems had a second prominent decoding peak (open arrowhead) following the initial peak (curved arrow; to facilitate comparison, the y-axis value of the open arrowhead is the height of the initial peak of the identity decoding problems). For the generalization problem, the second peak had the same or greater amplitude as the initial peak. We attribute the first peak to decoding the onset of the stimulus and the second peak to decoding the cessation of the stimulus.

**Decoding hue.** Figure 3b shows the average performance across the 12 hue identity problems (solid line) and the 12 generalizing-across-luminance-polarity problems (dashed line). The plot shows that hue was decodable in both cases. The fact that it was possible to decode hue generalizing across luminance polarity supports the hypothesis that the brain has a representation of hue that is separate from the representation of luminance contrast. The latency at which decoding became significant was 115 ms [35, 355] for the generalization problems, and 80 ms [75, 85] for the identity problems (the latency of the generalization problems was greater than the upper 95% CI limit for the latency of the identity problems, but a direct comparison of the bootstrapped values was not significant, $p = 0.09$). As with the luminance-polarity decoding problems, decoding had a higher peak magnitude for the identity problems (78% [74, 82]) compared to the generalization problems (56% [53, 59]), which provides support for the hypothesis that the brain also has a representation of hue that is inseparable from the representation of luminance contrast. But importantly, decoding of hue was substantially more impacted by changes in luminance polarity, than decoding of luminance polarity was by changes in hue. Nonetheless, the time to peak was not different for the identity and generalization problems (solid

vertical line, 121.8 ms [120, 130]; dashed vertical line, 119.1 ms [115, 130]; $p = 0.82$). Moreover, the time to peak was not significantly different for identity problems among luminance-contrast increments (122 ms [115, 130]) compared to identity problems among luminance-contrast decrements (123 ms [115, 130]); and the magnitude of peak decoding accuracy for these two sets of problems also were not significantly different (for light stimuli: 77% [73, 82]; for dark stimuli: 78% [74, 83]) (Supplementary Fig. 4). These results show that any potential difference in saturation between the light and dark stimuli does not influence the time course of decoding (saturation is ill-defined, but as discussed in the "Methods" section, there is an argument that the saturation of the dark stimuli was higher than the saturation of the light stimuli). The results of Supplementary Fig. 4 also support the argument that the differences in timing of decoding hue and luminance polarity cannot be attributed to subtle differences in the contrast of the hue and luminance-polarity stimuli. Finally, to the extent the solid line in Fig. 3b shows a second peak (open arrowhead), it was less pronounced than the initial decoding peak (curved arrow) (as in Fig. 3a, the y-axis value of the open arrowhead is at the height of the initial peak for the identity decoding problem).

**Comparing the temporal dynamics of decoding hue and luminance polarity.** To directly compare the timing of hue and luminance-polarity problems, we subsampled the data to obtain equal peak decoding across the two sets of problems[63] (the height of the initial decoding peaks of the blue and black traces are matched in Fig. 4a, b). We note that the use of subsampling only accounts for differences in the magnitude of peak decoding on the onset of decoding accuracy, but other factors that could cause differences in decoding onsets, such as trial temporal variability,

may not be accounted for. But importantly, the main conclusions about differences in timing of the representations of hue and luminance-contrast polarity are based on the time-to-peak decoding, not on the onset of significance of decoding, and the time-to-peak decoding did not change following subsampling.

The times to peak for the subsampled luminance-polarity data were 99.8 ms [95, 110] for the generalization problem and 103.4 ms [100, 110] for the identity problem. The comparison of the hue and luminance-polarity identity problems (Fig. 4a) and of the hue and luminance-polarity generalization problems (Fig. 4b) underscores three differences in time course of decoding. First, hue was decodable after luminance polarity as assessed by the time of peak decoding (identity problems: 18 ms delay, $p = 0.003$; generalization problems: 19 ms delay, $p = 0.017$). Differences in the latency of decoding onset showed a trend that reflects these timing differences, but the differences in latency of decoding onset were not significant (for identity problems, $p = 0.13$; for generalization problems, $p = 0.09$). Second, the time of peak hue decoding corresponded to the dip in the luminance-polarity decoding curve, and vice versa; the time point of peak luminance-polarity decoding corresponded to a notch in the hue-decoding curve. This was especially clear in the generalization curves, where the black and blue traces in Fig. 4b are in counterphase (quantified as the Spearman correlation of the derivative of the decoding curves, computed for 116 ms following the average of the onset latencies of the hue and luminance-polarity generalization problems; $R = -0.53$, $p = 0.01$). Third, for the identity problems, the second peak (open arrowhead) was larger for luminance polarity than for hue (double-headed arrow, $p < 0.001$; note that precise $p$ values are stated throughout the paper except when the $p$ value must be smaller than $p = 0.001$. In those cases, because the $p$ values were derived from 1000 bootstrap iterations, the precise $p$ value is unknown and the $p$ value is listed as <0.001).

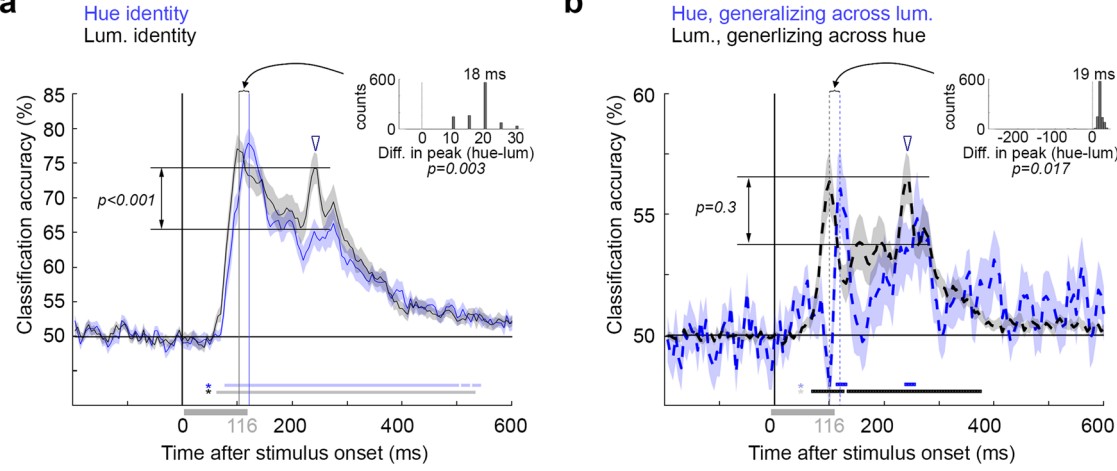

**Fig. 4 Comparing the temporal dynamics of decoding luminance polarity versus hue. a** Classification accuracy for the identity problems (see Fig. 3a, c). The data used to decode luminance identity was subsampled so that the maximum classification accuracy for luminance identity was the same as that of hue identity (61% of the trials were used). With the subsampled data, luminance polarity decoding peaked at 103.4 ms [100, 110], which is not different from the time of peak decoding of the complete data set (Fig. 3a). Inset shows the differences between the peaks across the 1000 bootstrapped samples across the 18 participants. Accuracy for luminance polarity peaked 18 ms before hue ($p = 0.003$, one-sided test comparing the bootstrapped distribution of the times to peak; the $p$ value indicates that three of the 1000 bootstrapped times to peak of the hue problems were earlier or equal to the bootstrapped times to peak of the luminance problems.). The magnitude of classifier accuracy corresponding to the cessation of the stimulus (open arrowhead) was greater for luminance polarity than hue ($p < 0.001$; exact $p$ values not possible, see "Methods" section; not adjusted for multiple comparisons; $p$ value indicates that none of the 1000 bootstrapped decoding problems showed higher classification accuracy at stimulus cessation for hue compared to luminance polarity). Shading shows the SE of the bootstrap samples. **b** Classification accuracy for the generalization problems. The data used to decode the generalizing-luminance-polarity problem was subsampled to match the initial peak of generalizing hue (16% of the trials were used). The subsampled generalizing-luminance-polarity problem peaked at 99.8 ms [95, 110], which is not different from the time of peak decoding for the complete data set (Fig. 3a). Accuracy for luminance polarity decoding peaked 19 ms before hue ($p = 0.017$). The magnitude of classifier accuracy corresponding to the cessation of the stimulus was not greater for luminance polarity than hue ($p = 0.3$). $p$-values in **b** computed in the same way as for panel **a**.

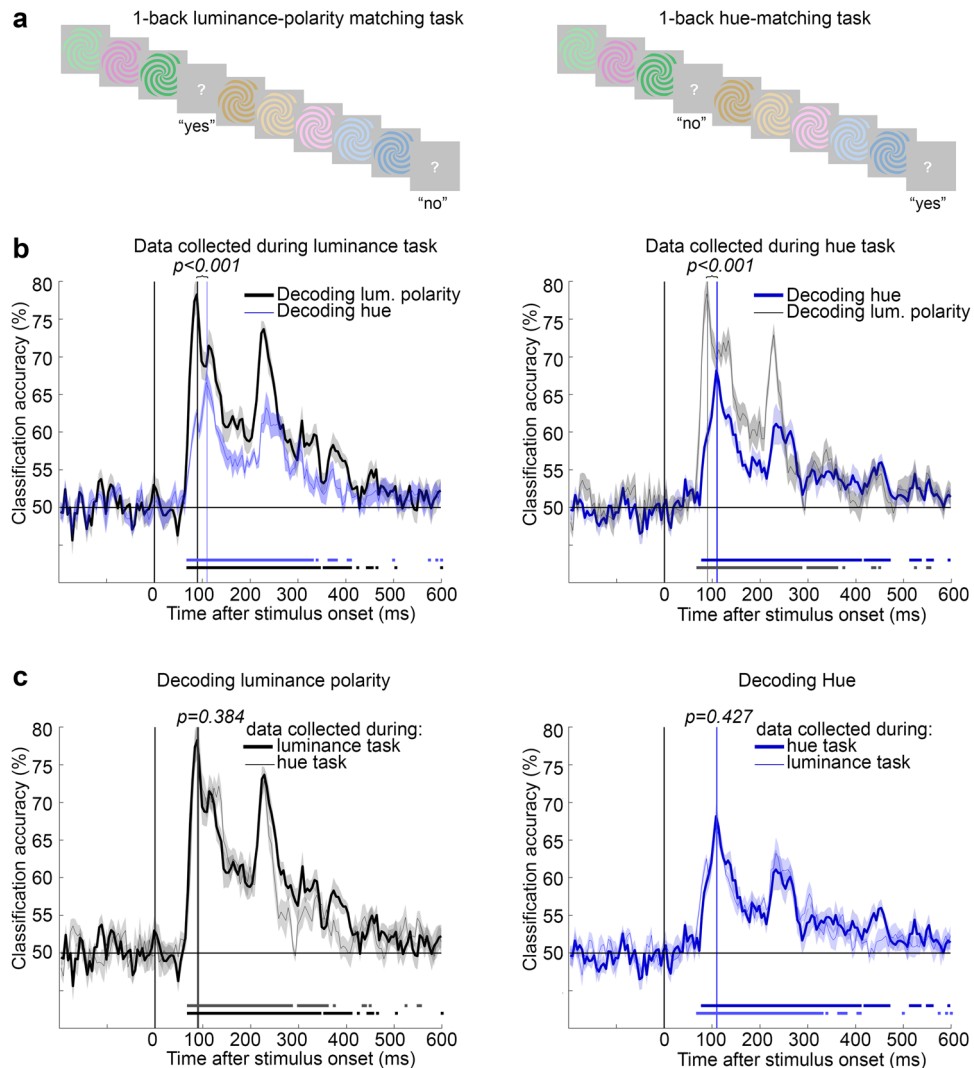

**Fig. 5 Task had no impact on the time to peak decoding of hue or luminance polarity. a** Tasks, 1-back luminance-polarity matching (left), 1-back hue matching task (right); note that the sequence of stimuli illustrated is the same for both tasks but the correct answers are different. **b** Left panel, Decoding luminance polarity (black trace) and hue (blue trace) using data from the luminance-polarity matching task. The data show the average of the identity decoding problems (see Fig. 1). Traces represent the average of 1000 bootstrapped time courses across eight sessions (two participants, four sessions each). Shading is SEM of the bootstrapped time courses. Vertical lines show the time to peak. The horizontal sequence of points above the x-axis show when decoding was greater than chance for four or more consecutive time bins (determined using within-subjects null decoding distribution and FDR corrected, see "Methods" section). Hue decoding peaked after luminance-polarity decoding ($p < 0.001$). Right panel, Decoding luminance polarity (black line) and hue (blue line) using data from the hue-matching task. Hue decoding peaked after luminance-polarity decoding ($p < 0.001$). Other conventions as for the left panel. **c** Left panel, Decoding luminance polarity using data from the luminance-polarity matching task (bold line) and the hue matching task (thin line). Shading represents the SEM. The time-to-peak for decoding luminance polarity identity was not impacted by task ($p = 0.384$). Right panel, Decoding hue using data from the hue matching task (bold line) and luminance-polarity matching task (thin line); the time-to-peak for decoding hue was not impacted by task ($p = 0.427$; $p$ values reflect two-sided $t$-tests using a bootstrapping procedure, not adjusted for multiple comparisons). In all panels, the bolded traces show the decoding problems that align with the task. Luminance-polarity decoding was significant for a longer duration for data collected during the luminance-polarity matching task (left panel), while hue decoding was significant for a longer duration for data collected during the hue matching task (right panel).

The same trend was observed for the generalization problems, but it was not significant ($p = 0.3$). Fig. 4 shows that for both the identity and generalization problems, the second peak was temporally more precise for decoding luminance polarity than for decoding hue (the black traces have a narrower, tighter second peak compared to the blue traces).

**Impact of task on decoding.** Decoding accuracy of visual stimuli from patterns of MEG activity could be impacted by task engagement during MEG data collection[64,65]. Task effects can enhance task-relevant object features and reportedly arise

relatively late after stimulus onset[66,67]; nonetheless, we asked whether the differences in timing for decoding hue versus luminance polarity could be attributed to the task that participants performed during MEG data collection. We answered this question in a control experiment conducted prior to the main experiments. The control experiment was identical to the main experiments except that for half the runs, the two participants performed a 1-back hue-matching task, and for the other half of the runs they performed a 1-back luminance-polarity matching task (Fig. 5a; each participant completed five sessions, one of which was not analyzed because of a technical error in data acquisition; each trial involved a 116 ms stimulus presentation

and a 1 s ISI; the task alternated between runs in each session). Participants learned both tasks, achieving excellent accuracy within the first MEG data-acquisition session and improving in reaction time over consecutive sessions (Supplementary Fig. 5).

The time to peak decoding was quantified by training and testing classifiers for each session and each participant (4 sessions × 2 participants for each task type) and bootstrapping across the eight curves for a given task. Peak decoding times were different for hue than for luminance polarity, for data collected using both tasks: decoding of hue peaked later than decoding of luminance polarity (Fig. 5b; difference in time to peak for data obtained using the 1-back luminance task, $p < 0.001$; difference in time to peak for data obtained using the 1-back hue task, $p < 0.001$). The time to peak for decoding hue was 112 ms [106, 119] using data obtained with the hue task, and 111 ms [101, 121] using data obtained with the luminance-polarity task; while the time to peak for decoding luminance polarity was 89 ms [84, 95] for the hue task, and 89 ms [81, 101] for the luminance-polarity task. These results replicate the results from the main experiments, showing that decoding luminance polarity reached a peak about 20 ms earlier than decoding hue. The time to peak decoding of luminance polarity was indistinguishable for data collected under the two task conditions (Fig. 5c, left panel; $p = 0.384$). Similarly, the time to peak decoding of hue was indistinguishable for data collected under the two task conditions (Fig. 5c, right panel; $p = 0.427$). These results show that task did not impact the time to peak, for decoding hue or luminance polarity. The data collected in the control experiment also show that the relative timing difference for decoding hue versus luminance polarity was apparent for each individual subject: combining data from both tasks, the time to peak for decoding hue for subject 1 was 108 ms [101, 117], and for subject 2 was 114 ms [110, 118]); the time to peak for decoding luminance polarity for subject 1 was 88 ms [86, 90], and for subject 2 was 92 ms [84, 102]). The time to peak for

decoding hue was different from the time to peak for decoding luminance polarity in both subjects (both S1 and S2, $p = 0.001$).

The results of the control experiment provide evidence of one possible impact of task: the duration of significant decoding. Decoding of luminance polarity was significant for a longer duration when participants were engaged in the 1-back luminance-polarity task compared to when they were engaged in the 1-back hue task (Fig. 5c, left panel, horizontal lines below the decoding trace show time points of significant decoding). Similarly, decoding of hue was significant for a longer duration when participants were engaged in the 1-back hue task compared to when they were engaged in the 1-back luminance-polarity task (Fig. 5c, right panel). The results of the control experiment are consistent with the manifestation of task-related effects arising relatively late (>300 ms after stimulus onset), as documented by others[66], and argue against the hypothesis that the difference in time-to-peak for decoding hue and luminance polarity, which is evident much earlier than 300 ms, be attributed to task.

**Cross-temporal decoding for hue and luminance polarity.** The results discussed so far evaluate the classifiers' performance using test data obtained at the same time point after stimulus onset as the data used to train the classifier. The classifiers show significant decoding for a substantial amount of time, hundreds of milli-seconds, following stimulus onset. One possibility is that the pattern of activity is relativity sustained over this time period; another possibility is that it is dynamic and transient[68]. To distinguish between these alternatives, we trained classifiers on the patterns of activity at each point in time and evaluated the extent to which they could predict activity at all other time points. If activity patterns are dynamic, the cross-temporal analysis will recover strong decoding performance only for situations in which the training and testing data sets were obtained at the same timepoint relative to stimulus onset, i.e., along the diagonal in a

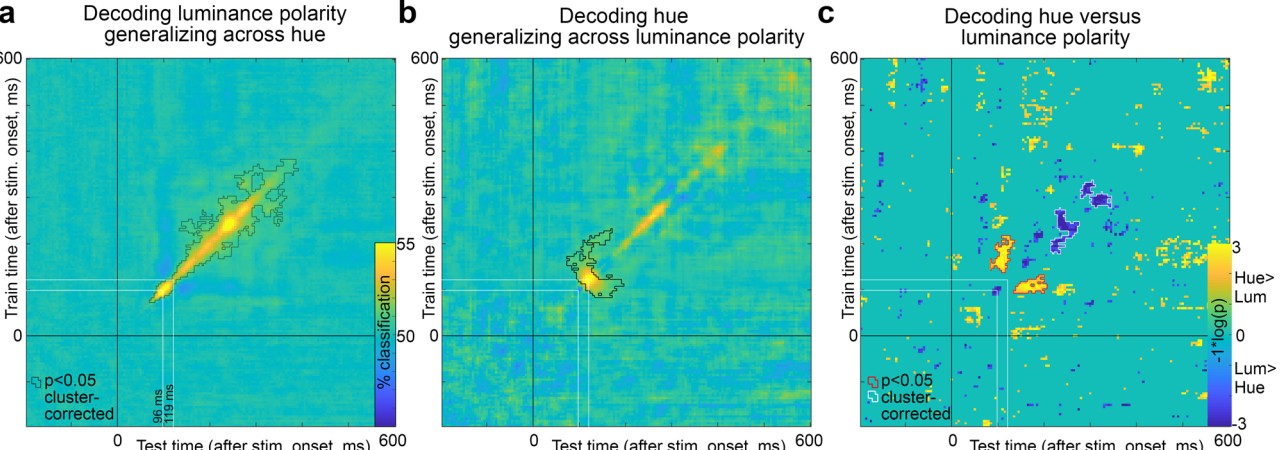

**Fig. 6 Cross-temporal generalization of decoding luminance polarity versus hue. a** Classifiers were trained using the pattern of MEG activity elicited at time points from −200 ms to 600 ms after stimulus onset (y-axis) and tested using data not used in training across the same time interval, on the set of problems decoding luminance polarity generalizing across hue. The best decoding performance was achieved with classifiers that were trained and tested using data from the same time point after stimulus onset, indicated by the strong performance along the $x = y$ diagonal. The peak classification time was at 100 ms; there was a dip in classification performance at about 120 ms. The black contours show regions in the heatmap that were $p < 0.05$ cluster corrected. The $p$ values were obtained using a sign-permutation test. **b** Data as in **a**, but for classifiers trained and tested on the set of problems decoding hue generalizing across luminance polarity. The peak classification time was 119 ms; the cluster-corrected regions of significance extended further from the diagonal compared to **a**. **c** Comparison of results in **a** and **b**. The time points where the classifiers were more accurate for luminance polarity compared to hue are shown as dark blue (white contours show cluster-corrected significant results), while the time points where the classifiers were more accurate for hue compared to luminance polarity are shown as yellow (red contours show cluster-corrected results). The $p$ values were obtained using a permutation test on the difference in accuracy between hue and luminance polarity decoding. The $p$ values are cluster corrected. Decoding of hue showed greater generalization across time compared to decoding of luminance polarity, and the greater cross-temporal generalization for hue began relatively early after stimulus onset (~119 ms).

cross-temporal decoding plot. Alternatively, if activity patterns are sustained, the analysis will show strong decoding performance at time points away from the diagonal.

Figure 6 shows the cross-temporal plots for decoding luminance polarity generalizing across hues (Fig. 6a, left), and for decoding hue generalizing across luminance polarity (Fig. 6b, right); the figures show the results averaged over the 12 individual problems for luminance polarity and hue using the subsampled data shown in Fig. 4, which ensures that any differences in cross-temporal decoding cannot be not attributed to absolute differences in peak decoding. The color scale in the heatmap shows the percent classification—the values along the diagonal are the same as those shown in Fig. 4b. The black contours identify data that were deemed significant by a permutation test (significance threshold $p < 0.05$) and cluster correction (cluster defining threshold $p < 0.01$, see "Methods" section). The cluster correction in two dimensions did not recover as significant all the same values along the diagonal that were deemed significant by the cluster correction in one dimension in Fig. 4b because it is more stringent. Despite the greater stringency, the cross-temporal generalization for decoding hue showed greater significance further away from the diagonal compared to the cross-temporal generalization for luminance polarity (compare Fig. 6a, b). Figure 6c quantifies this comparison: yellow regions in the plot indicate where hue decoding was greater than luminance-polarity decoding, while dark blue regions indicate where luminance-polarity decoding was greater than hue decoding (the comparison was determined through a permutation test on the difference between luminance-polarity and hue decoding accuracy; red and white contours show cluster-corrected results). The relatively stronger cross-temporal generalization for hue, indicated in the plot as the yellow flanks around the diagonal, began relatively quickly after stimulus onset, emerging at about the same time as the initial peak in hue generalization decoding (119 ms). The cross-temporal decoding plots also show evidence for faint bands of decoding parallel to the diagonal corresponding to a reactivation time of ~50 ms, as described previously[26].

These results support the hypothesis that the patterns of activity in the brain associated with hue are more sustained than the patterns of activity associated with luminance polarity.

**Luminance-polarity decoding with different combinations of hue for training and testing**. The decoding analysis in Fig. 3 shows that luminance polarity can be decoded from the pattern of MEG data but does not address the variability in the extent to which luminance-polarity information can be decoded using different combinations of hues for training and testing the classifiers. We were interested in addressing this question because the extent to which the brain relies on both L-M and S cone mechanisms for encoding luminance contrast is unclear[4–6], and because some behavioral data suggest that luminance is less reliably extracted from colors associated with the daylight locus (orange/blue) compared to the anti-daylight locus (pink/green)[69–74]. For example, the chromaticity of natural lights is linked to relative luminance contrast: direct sunlight has a warm spectrum (e.g., orange), and is associated with a luminance increment, whereas indirect light, such as in shadows, has a cool spectrum (reflecting skylight) and is associated with a luminance decrement. If the visual system is adapted to natural statistics, we predicted that luminance polarity carried by orange and blue would be less meaningful about object boundaries than luminance polarity carried by green and pink, because luminance contrasts linked to orange and blue chromaticity are more likely to be attributed to the illumination.

Individual luminance-polarity decoding problems averaged over participants were variable both among the four identity problems (Fig. 7a) and among the twelve generalization problems

(Fig. 7b). The color scale of the heatmap in Fig. 7c indicates the luminance-polarity decoding performance for each of the 16 subproblems during the 5 ms time bin at the time of the initial peak decoding determined separately for each subproblem (these results are similar to those obtained during the 5 ms time bin set by the average peak decoding for the luminance-polarity identity problems). Data along the inverse diagonal correspond to the identity problems (recall the identity problems are those in which the hue was the same for training data and testing data). Data off the diagonal correspond to decoding problems for luminance polarity generalizing across hue, and these data allow us to test ideas about how S and L-M signals contribute to the representation of luminance polarity. The time bin of peak decoding for each problem is indicated in each entry. The bolded numbers are the total number of 5 ms time bins in which decoding was significant (cluster-corrected, cluster-defining threshold $p < 0.01$). Luminance polarity could be decoded above chance with data obtained in all combinations of training and test hues (significance of decoding was determined by a permutation test, see "Methods" section), but there was variability in the magnitude and duration of decoding depending on the hue used to obtain data for training versus testing the classifiers (the 95% CI of peak decoding accuracy values obtained by bootstrapping are shown in square brackets).

The data in Fig. 7 support three findings. First, luminance-polarity decoding was strongest when the carrier (training) hue was the same as the test hue (paired two-sided Wilcoxon signed-rank test on average peak decoding values comparing on-diagonal to off-diagonal Fig. 7c, $p = 0.0002$). Second, among these identity problems, classification performance was higher for warm colors (pink, orange) than cool colors (blue, green; repeated measures one-way ANOVA, Fig. 7c, $p = 0.003$; supporting our prior analysis[26]). Third, for a given pair of hues used to obtain the training and testing data, the performance of the classifiers was consistent regardless of which data set was used for training and which was used for testing; this is reflected in a similar pattern in the upper and lower triangles of the heat map formed across the inverse diagonal in Fig. 7c. For example, the accuracy for training on orange, testing on pink, is comparable to the accuracy for training on pink, testing on orange. But the magnitude of the classification accuracy and the number of significant time points recovered by the classifier varied with different combinations of hues for training/testing; for example, training (or testing) with orange and testing (or training) with pink yielded different results than training (or testing) with orange and testing (or training) with blue. We had hypothesized that classifiers trained and tested using hues associated with the daylight axis (blue and orange) would be less successful at decoding luminance polarity than classifiers trained and tested using hues associated with the anti-daylight axis (pink and green). The results show a trend supporting this hypothesis (Fig. 7c), but it was not significant (repeated measures one-way ANOVA, Fig. 7c, $p = 0.076$).

Perhaps the appropriate test of our hypothesis should examine not the magnitude of decoding but rather the duration over which each problem could be decoded. The number of 5-ms time bins that showed significant decoding was lower for problems involving hues of the daylight axis (34 or 36 time bins, bolded numbers, Fig. 7c) compared to problems involving hues of the anti-daylight axis (42 or 45 bins), but this difference was not significant (Fig. 7d shows the 95% CI of these values). Instead the prominent result of these analyses is that classifiers trained and tested with hues that differ in both their L-M and S cone modulation were worse than classifiers trained and tested with hues that differ in modulation of only one cone system (L-M or S). Figure 7d shows the results on the cone-opponent axes that define the color space, in which the color of the letters and arrows

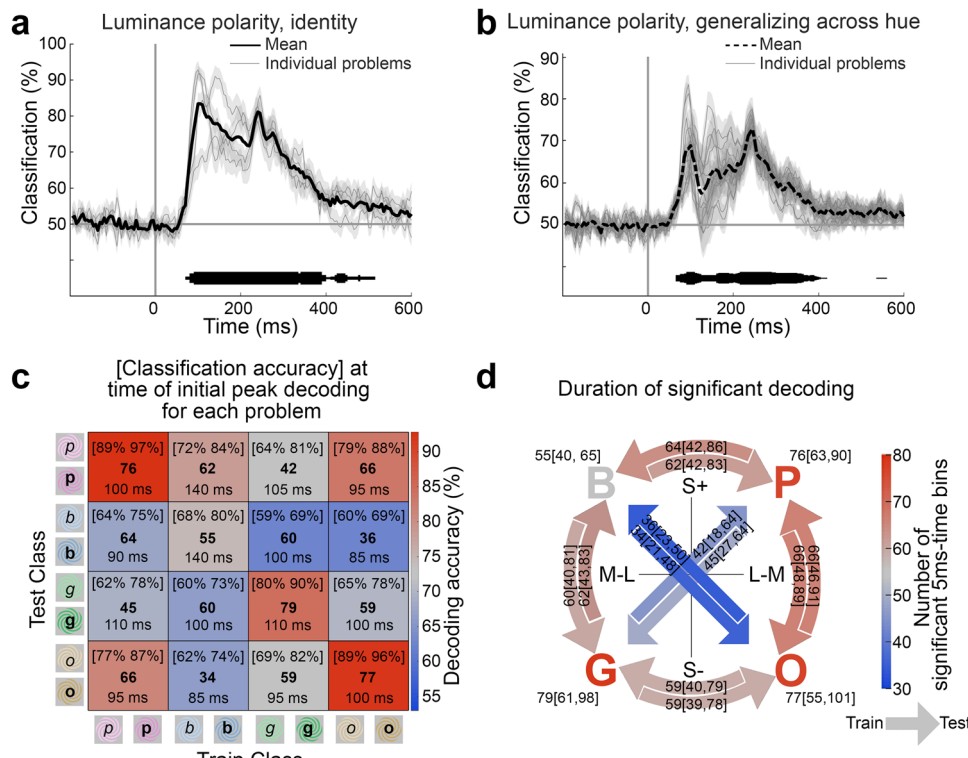

**Fig. 7 Decoding luminance polarity carried by different hues. a** Accuracy of classifiers trained to decode luminance polarity on the identity problems (individual problems, thin lines; average, thick line). **b** Accuracy of classifiers trained to decode luminance polarity across hues (12 generalizing-across-hue problems, thin lines; average, thick dashed line; shading shows SE across subjects). The thickness of the horizontal line above the *x*-axis shows the number of decoding problems (maximum in **a,** 4; maximum in **b,** 12) that were significant in each 5 ms bin (cluster corrected). **c** Heatmap showing classifier performance [95% CI, by bootstrapping] for each binary classifier at the initial peak decoding for each problem. Bold numbers show the total number of 5 ms time bins with significant classification performance. The time stamps show when the classifier performance reached the initial peak in decoding accuracy. **d** Duration of significant decoding of luminance polarity plotted in the cone-opponent color space used to define the stimuli; B, P, O, and G demarcate within this space the location of the blue, pink, orange, and green hues of the stimuli. The color scalebar indicates the number of 5 ms time bins that were significant, and the arrow direction points from the hues used for training to the hues used for testing. For example, the arrow from P to G corresponds to classifiers trained on luminance polarity using data obtained with light/dark pink stimuli and tested using data obtained with light/dark green stimuli. For the four identity problems (the letters B, P, O, and G), the hue was the same for training and testing. For each type of problem, the precise number of significant time bins is shown, along with the [95% CI] obtained by bootstrapping. Problems that cross the color space, in which the hues for training and testing varied in both L-M and S activation, showed lower numbers of significant time bins than problems around the perimeter of the color space, in which the hues for training and testing varied in only L-M or S.

corresponds to the number of significant time bins for each problem (P, B, G, and O correspond to the four hues of the spirals: pink, blue, green, and orange; each problem is labeled with number of significant time bins and the 95% CI obtained by bootstrapping). The direction of the arrows indicates which data was used to train and test the classifiers; for example, the arrow from P to B shows the number of time bins in which the luminance polarity problems were significant when training on data obtained using light and dark pink, and tested on data obtained using light and dark blue. The arrows connecting hues through the color space, in which the hues for training and testing differ in both L-M and S modulation, showed a lower number of significant time bins compared to problems around the color circle, in which the hues for training and testing differ in only one or the other cone-opponent mechanism (the arrows through the origin of the space are bluish, while all the problems around the perimeter are pink or gray). Thus the results do not provide strong support for our initial hypothesis that neural mechanisms encoding luminance polarity are strongly impacted when the hues of the stimuli correspond to the daylight locus, but they do support the idea that luminance polarity is encoded by a combination of both L-M and S chromatic mechanisms.

The statistical analysis shown in Fig. 7 was obtained by comparing and averaging results across participants, but the time course of decoding accuracy may show individual differences that will not be apparent in these averages. One possibility we considered is whether the somewhat weaker luminance-polarity decoding for blue and orange reflects variability in the timing and amplitude of decoding across participants. To address this possibility, we first determined the average decoding accuracy throughout the decoding time course (0–600 ms) for each problem, for each participant (72 problems total: 18 participants × 2 daylight-axis problems × 2 anti-daylight-axis problems). The average peak decoding accuracy after stimulus onset was not different for daylight colors and anti-daylight colors (55% [53.8, 57.0] versus 56% [54.1, 57.8], $p = 0.68$, repeated measures ANOVA). Next, for each problem, we calculated the Spearman correlation of the decoding time course after stimulus onset for each of the 18 participants with the average time course of a sample of 17 subjects drawn with replacement from the other participants, to estimate the temporal correlation of the decoding curves. We repeated this 1000 times and averaged across the repetitions to obtain a mean estimate of the temporal correlation of each participant's time course to those of the others. If the time

course of decoding shows the same structure across participants, the analysis will yield high temporal correlations. The average correlation for problems across the daylight colors versus across anti-daylight colors was not different (0.33 versus 0.30; $p = 0.62$ repeated measures ANOVA). Problems involving daylight colors were not different from problems involving anti-daylight colors in both their mean accuracy and temporal correlation (MANOVA, $p = 0.37$; note that the correlation of the average decoding accuracy with the temporal correlation, Supplementary Fig. 6, is not surprising, because when decoding is higher, the temporal correlation will be more apparent).

**fMRI-guided MEG source localization**. We were interested in evaluating the extent to which MEG signals arising from functionally defined regions in the cortex could support decoding of hue and luminance polarity. To address this question, we ran fMRI experiments in 14 of the same participants in whom we collected MEG data, and we performed the MEG analyses using subject-specific source localization. Our goal was to use functional data to define regions of interest in the ventral visual pathway (VVP) in each participant, controlling for individual differences in the absolute location of functional domains across people. In each subject we used fMRI to identify regions biased for faces, places, colors, and objects, using a paradigm we previously implemented in which fMRI responses to short movie clips of faces, bodies, objects, and scenes were measured[75,76]. The paradigm involved measuring responses to intact and scrambled versions of the clips, and to clips in full color and black-and-white. As described in Lafer-Sousa et al.[76], the results can be used to define regions of interest: face-biased regions (including the FFA); place-biased regions (including the PPA); and color-biased regions. The results also recover area LO (lateral object area), defined by stronger responses to intact versus scrambled movie clips.

Figure 8a shows the fMRI results for one participant: greater responses to colored movie clips compared to black-and-white versions of the movie clips is shown by the heat map, apparent on the ventral view (Fig. 8a, right panel); functional domains for faces (faces>objects), objects (intact objects>scrambled objects), and places (places>objects) are indicated by contour maps drawn at $p = 0.001$ threshold. Color-biased activity was found sandwiched between place-biased activity (medially) and face-biased activity (laterally), confirming prior observations[76]. By aligning each participant to a standard atlas[77,78], we also generated regions of interest for V1, V2, and MT, and for frontal cortex and the precentral gyrus (control regions).

MEG signals source-localized to V1 and V2 yielded the highest magnitude modulation in current source density averaged across all stimulus presentations (Fig. 8b). The magnitude of the CSD was different among the functional regions identified in the VVP ($p = 0.002$, repeated measures one-way ANOVA): the color-biased regions were not different from face-biased regions ($p = 0.12$; paired two sided $t$-test, uncorrected); but were different from LO ($p = 0.01$), and from place-biased regions ($p = 0.02$) (Fig. 8c). These results provide a direct measure of neural activity, and confirm the indirect measurements obtained with fMRI suggesting that fMRI-identified color-biased regions (and possibly face-biased regions) play an important role in color processing[79].

Luminance polarity generalized across hue was decodable to some extent in all visual regions except the face-biased regions and the color-biased regions; it was most decodable in V1 and V2; to a lesser extent in MT and LO; and to an even lesser extent in the place-biased regions; it was not decodable in the two control regions (Fig. 8d). Hue generalized across luminance polarity was not reliably decodable in any region, but with a hint of decoding

in V1 and/or V2 (Fig. 8e). The distribution of sensors used in the decoding analysis is shown in Fig. 8f, g (see legend and methods for details).

## Discussion

The experiments here produced five results: first, hue and luminance polarity could be decoded independently. Luminance polarity could be decoded even if the training and testing stimuli differed in hue; and hue could be decoded even if the training and testing stimuli differed in luminance polarity. Second, classification accuracy was higher for problems in which the stimuli used to train and test the classifiers were identical (identity problems, Fig. 3), compared to problems that required generalization across hue or luminance polarity (generalizing problems, Fig. 3). Third, the time course of decoding was different for hue compared to luminance polarity (Fig. 4). The peak for decoding hue was ~20 ms after the peak for decoding luminance polarity. Control experiments showed that these timing differences could not be attributed to the task that participants performed during MEG data collection (Fig. 5). Fourth, representations of hue showed subtly greater cross-temporal generalization than representations of luminance polarity (Fig. 6). And fifth, representations of luminance polarity varied depending on the hues used to obtain the training and testing data (Fig. 7). Decoding of luminance polarity was most reduced when the hues used to obtain the training and testing data varied in both L-M and S, suggesting that both cone-opponent retinal mechanisms contribute to luminance encoding. Together, the results have implications for our understanding of the different roles that luminance polarity and hue play in visual perception, and how encoding these features is implemented by the neural circuitry.

Multivariate analyses of MEG and EEG data uncover important information about color processing in the brain[26,80–85]. We interpret the decoding results as evidence of the temporal evolution of the neural representations in the brain, an interpretation we will return to in the discussion below. But first we are prompted to ask whether the recognized fine temporal structure of MEG (and EEG data) is sufficient to explain the decoding analyses without requiring information about the spatial pattern of MEG signals across sensors. For example, two stimuli that elicit the same temporal profile of response but with slightly different latencies, will be decodable from a single electrode (so long as the response is not uniform across time), because at any given time point there would be a predictable difference in the amplitude of the signals. Is temporal structure sufficient to explain the results, or do the results imply that there is a spatial representation of color that gives rise to distinct spatial activation patterns in response to different colors? Each analysis, e.g., of hue identity, depends on averaging many separate decoding problems to extract the common decodable information across the spatial pattern of sensors at each time point. Such averaging would obscure any decoding dependent solely on latency differences, because the difference in univariate response time course would likely be different for any given pair of stimuli. For latency differences to be the sole explanation, the brain would need to respond with different timing to different hues, and the differences in timing between any pair of hues would need to be the same (and immune to changes in hue and/or luminance contrast) to yield the present results. These considerations raise the possibility that the decoding reflects differences in the spatial representation of the stimuli. Yet that explanation might be surprising given EEG and MEG are thought to show coarse spatial resolution, and the spatial scale of color columns, in early visual cortex[86–89] or in higher visual cortex[90], is relatively fine. The paradox could be resolved either if the spatial scale of color

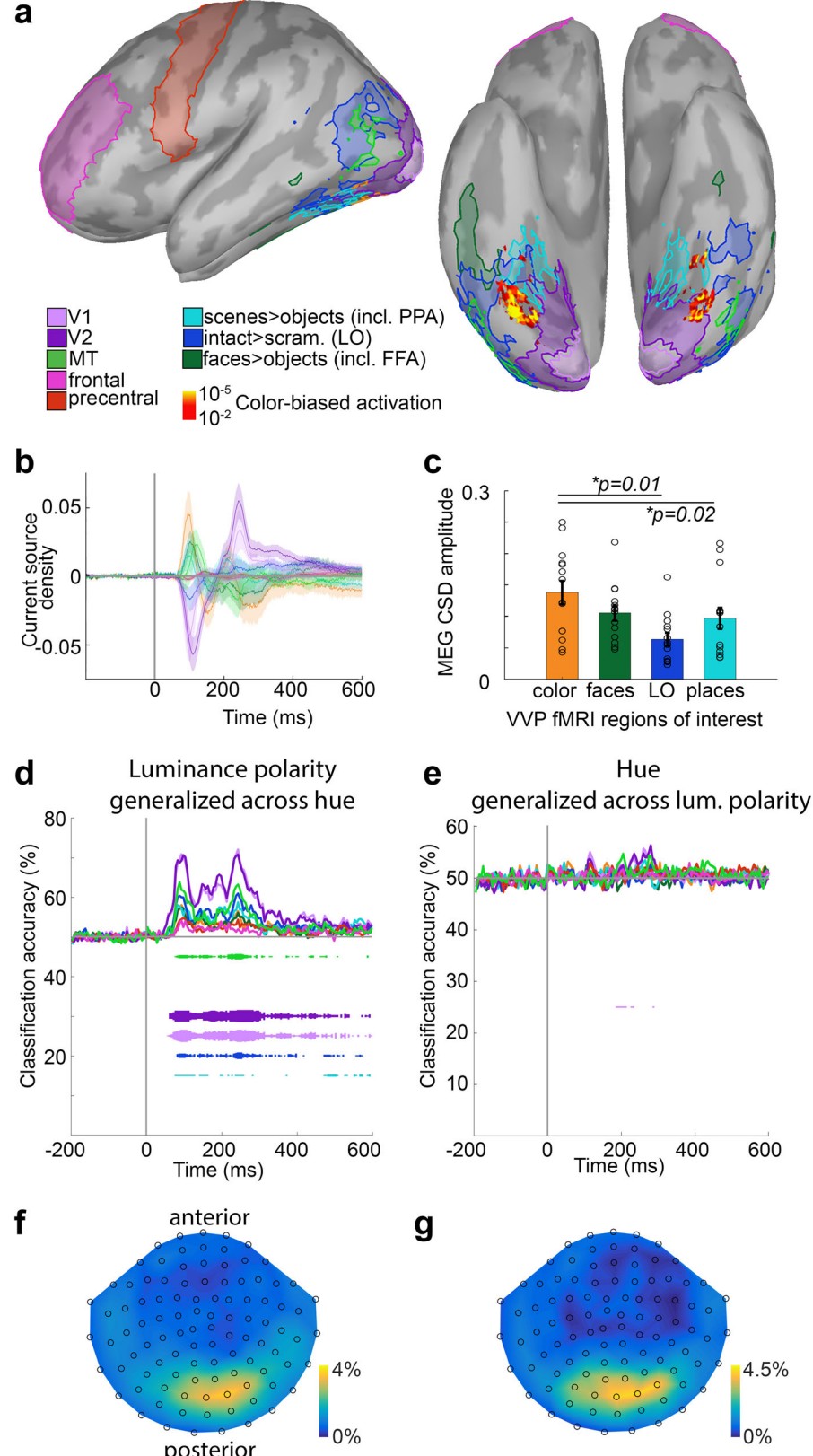

columns is coarser than currently estimated, or if the multivariate analyses are able to detect the existence of rather fine spatial patterns, even if the analyses cannot recover the precise spatial organization of them, as suggested by others[26,91].

The present evidence for significant decoding of hue generalizing across luminance polarity, and luminance polarity generalizing across hue, first presented in preliminary form[80,81], is consistent with results of a recent independent report[92]. Decoding of hue was substantially more impacted by changes in luminance polarity, than decoding of luminance polarity was by changes in hue. This evidence shows that either luminance contrast and hue are encoded to some extent by different neural

**Fig. 8 Source localization of regions defined using fMRI. a** Functional anatomy from one participant, on the inflated cortical surface (left panel, side view, anterior to left; right panel, ventral view, anterior at top; similar results were obtained in 14 participants). Regions of interest were defined using fMRI responses to movie clips, in color and black-and-white, of faces, bodies, objects, scrambled objects[76]. The activation map (yellow-orange) shows voxels with higher responses to color clips compared to black-and-white clips. Contours show ROIs for faces > objects (including the fusiform face area, FFA), intact shapes > scrambled shapes (lateral object area, LO), and places > objects (including parahippocampal place area, PPA). Regions of interest for V1, V2, MT (middle temporal area), frontal, and precentral ROIs were defined using an anatomical atlas. **b** Source localization estimates of the current source density (CSD) of magnetoencephalography (MEG) responses to color arising from each functional ROI, averaged across participants and calculated with dynamical Statistical Parametric Mapping (dSPM; see "Methods" section). 0 on the x-axis is stimulus onset. Values on the y-axis are unitless. Transparent shading shows SEM. **c** Maximum amplitude of CSD in each ROI of the ventral visual pathway, calculated as distance from peak to trough of the time course in **b**. Error bars are SEM ($n = 14$). There was a significant effect of ROI on response magnitude (repeated measures one-way ANOVA, $p = 0.002$). Responses source-localized to the color-biased regions were different from those of LO (two-sided paired t-test, $p = 0.01$) and place-biased regions ($p = 0.02$), but not face-biased ROIs ($p = 0.12$). **d** Average classifier performance on decoding luminance polarity generalizing-across-hue (12 problems averaged together; see Fig. 1a) trained using only those MEG data localized to the MRI-defined ROIs ($N = 14$ participants). Each line shows the average accuracy of one ROI-restricted classifier averaged across participants (color key in **a**). **e** Average classifier performance on decoding hue generalizing-across-luminance-polarity (12 problems averaged together; see Fig. 1b). **f** The distribution of sensors used as features for the classifiers across participants ($N = 18$). Color bar shows the percent likelihood that any given sensor was selected as a feature. **g** As in **f**, but for decoding hue generalizing across luminance contrast.

populations, or if they are encoded by entirely the same population, the encoding must involve temporal multiplexing. Either possibility argues against the notion that luminance polarity and hue are necessarily multiplexed simultaneously by the same neural population. The relatively earlier timing of the luminance-polarity representation is consistent with univariate analyses of electroencephalography data showing that the early onset P1 component is driven by luminance, and the subsequent N1 peak is delayed by about 20 ms when elicited by equiluminant stimuli[43,93,94].

The difference in the time to peak of decoding hue versus luminance polarity was not impacted by task. The lack of an impact of task is consistent with prior work showing that attentional tasks involving sequential stimuli do not impact the feed-forward flow of information[95], and task effects typically emerge 200–300 ms after stimulus onset[66,67]. Consistent with the late arrival of task effects, at times >300 ms we observed subtly higher decoding of luminance polarity compared to hue for data obtained with the 1-back luminance-polarity task; and subtly higher decoding of hue compared to luminance polarity for data obtained with the 1-back hue task (Fig. 5c).

In object perception, the decoding time course reflects the perceptual and categorical dissimilarity of stimuli, with more perceptually dissimilar stimuli (i.e., higher levels of abstraction) decodable later, and associated with computations performed by areas further along the visual-processing hierarchy[96–98]. (An increase in category abstraction is Doberman→dog→animal→animate.) One way of thinking about the relatively later decoding of hue, then, is that (1) hue discrimination involves greater perceptual dissimilarity (greater category abstraction) compared to luminance polarity; and (2) it is computed either by circuits downstream of those that compute luminance polarity or requires more recurrent processing. The time course for decoding hue (peak 118–119 ms; Fig. 3b) is comparable to that for decoding shape-independent object category[99] and face identity[100], operations that probably reflect activity in area LO and the fusiform face area (FFA). The decoding time course for hue is therefore consistent with the hypothesis that hue is computed at about the same distance from the retina along the visual processing hierarchy as LO and the FFA, which implicates the posterior color-biased region of the VVP, part of the V4 Complex[76,101–103]. Cells in this region are spatially organized according to hue and their hue selectivity is tolerant to changes in luminance polarity[39,104,105]. Another way of thinking about the results is that the relatively later decoding of hue depends on a more sluggish channel from the retina through the lateral

geniculate nucleus up to visual cortex. This hypothesis is consistent with the idea that parvocellular neurons underlie hue decoding, while magnocellular neurons underlie luminance-polarity decoding. Measurements of decoding in patients with cortical lesions that impact color perception may prove instructive in addressing these alternative explanations[106].

Neural representations related to object vision that emerge earliest, as determined by classifiers tested using the same images on which they were trained (i.e., not requiring perceptual dissimilarity), reflect the encoding of "low-level visual features"—decoding in these classifiers peaks early (<100 ms), and is attributed to operations implemented early in the visual-processing hierarchy, perhaps V1[99,107–109]. But it has not been clear what constitutes the low-level visual features that determine the early decoding performance. It is often implied that the low-level features consist of both oriented luminance-contrast edges and color, as implied by separate luminance-contrast edge filters and color filters in the earliest layer of convolutional neural networks[12]. But the timing discovered here suggests that (1) luminance contrast (and not hue) is the first feature to be encoded; and (2) luminance contrast is implemented at an earlier stage in the visual-processing hierarchy, before hue. The relative timing difference for decoding hue and luminance polarity may be relatively small—just tens of milliseconds—but it is not only robust, evident within individual subjects (Fig. 5), but also long enough to encompass 7–10 synapses given monosynaptic transmission latencies of ~2.5 ms in the visual pathway[110].

Knowledge of the timing of the neural events may provide clues to the different roles that luminance contrast and hue play in vision[111]. The visual system is confronted with a constant stream of retinal images, each of which is associated with a cascade of neural activity lasting hundreds of milliseconds[64]. How does the visual system parse this stream of information to encode new content? It must do so while retaining some representations for longer durations to enable recognition and memory. In any situation in which information is encoded in time, it is advantageous to have clear signals indicating the start and end of the code, such as in genetics where gene sequences are parsed by canonical start and stop codons. In object vision, dynamical systems modeling predicts the existence of observable update mechanisms that signal new content[112]. The luminance-polarity decoding time course showed clear peaks corresponding to both the onset and cessation of the stimulus (Figs. 3, 4), and limited cross-temporal generalization (Fig. 6a). The time course for decoding hue, meanwhile, showed a weaker secondary peak and relatively stronger cross-temporal generalization (Fig. 6b, c). This

pattern of results is consistent with the idea that the brain more likely uses luminance contrast, rather than hue, as the updating signal, to encode discrete events embedded in the constant stream of visual information.

What, then, does the timing for decoding hue suggest about its role in vision? People with normal color vision find it difficult to make heterochromatic brightness matches[27] (see Supplementary Fig. 5). Moreover, humans are more likely to retain visual memory for hue than for luminance contrast[17]. Thus hue is a more salient dimension than luminance contrast, and more sustained. The present results provide a plausible neural correlate for these behaviors. They may also help resolve a paradox in visual neurophysiology. In the earliest studies of V1 single-unit responses, Hubel and Wiesel remarked that "it was surprising to us, however, that the great majority of cells [in V1] could discriminate precisely the orientation or direction of movement of a stimulus, but had no marked selectivity regarding wavelength"[113]. The seemingly weaker response of V1 cells to color compared to luminance contrast has fueled a lot of research since this early observation[114]. The present results open the possibility that hue is encoded by more sustained patterns of response to color across the population, which would not necessarily be evident using traditional single-cell recording methods.

Despite the simplicity of conventional representations of color space, which depict luminance contrast as an orthogonal to hue, substantial psychophysical work shows these dimensions interact (see "Introduction" section). Decoding luminance polarity was most reduced when the hues for training and testing varied in both L-M and S (the diagonals in Fig. 7d), suggesting that the brain encodes luminance polarity with both S and L-M mechanisms, which informs a long-standing discussion about which retinal mechanisms support the perception of luminance contrast[4–7].

Interactions between luminance and color might reflect an adaptation to natural lighting conditions. The chromaticity of natural lights covaries with relative luminance contrast, ranging from warm and bright (e.g., direct sunlight) to cool and dark (e.g., shadow). Luminance contrast signals associated with blue and/or orange chromaticities could be less reliable indicators of object boundaries since they may arise from boundaries introduced by cast shadows. The experiments were designed to test the hypothesis that luminance-contrast polarity is less readily encoded by the brain when carried by hues that align with the daylight axis (orange/blue) versus when carried by colors that do not (pink/green)[71–73,115]. Prior physiological work has examined the interaction of hue and luminance; this work has often tested colors aligned with the cardinal cone-opponent mechanisms (S and L-M)[5,7,43], which is insufficient to test the hypothesis since the daylight locus does not fall along the cardinal axes. If luminance representations are impacted by the S-cone mechanism, then the impact should be comparable for colors that have the same sign and magnitude of S-cone modulation, even if the L-M contribution differs. Alternatively, if luminance representations reflect an interaction with the daylight locus, the results should show an asymmetry in luminance polarity decoding that reflects an additional interaction of S and L-M. The results were consistent with this second prediction: decoding of luminance-contrast polarity was strikingly different for +S + (M-L) versus +S + (L-M) stimuli (these stimuli appear bluish and pinkish, are both S increments yet differ in the sign of L-M modulation). The results showed comparable luminance-polarity decoding accuracy for -S + (M-L) and -S + (L-M). Note that these results are not explained by the well-documented asymmetry in neural responses to S-increments versus S-decrements[116], which would predict similar decoding for S-increments regardless of the contribution of the sign of L-M. Thus the results show that one pole of the

daylight axis is compromised in encoding luminance contrast, the pole associated with the +S + (M-L) axis, which may relate to asymmetries in blue-yellow color perception[72]. We do not interpret the present data as evidence of neural adaptation to the daylight locus rather than as correlates of perception; instead, we argue that perception reflects adaptation to the daylight locus as implemented by neural representations.

There are substantial gaps in knowledge regarding the connection between fMRI and neural events. We leveraged the comparitively high spatial resolution of fMRI and the more direct access to neural events of MEG by obtaining both fMRI and MEG in the same participants. We used source localization to estimate MEG signals arising from fMRI-identified regions defined in each individual subject (Fig. 8). The results provide a way of independently testing conclusions drawn from fMRI experiments. For example, within the VVP, cortical regions showing the strongest fMRI responses to color are sandwiched between more lateral regions responding most strongly to faces and more medial regions responding most strongly to places[76]; this pattern was described earlier in macaque monkeys[101]. Among regions of the VVP, the MEG signals assigned to color-biased regions showed the largest current-source density in response to the stimuli used in the MEG experiments—these stimuli differed only in color and not shape. Thus the present results support the idea that the VVP comprises parallel streams characterized by differential sensitivity to color information[41]. Source-localized analyses did not recover significant luminance-polarity-invariant hue representations, which may not be surprising given fundamental limitations of source localization[117]. In future work, we intend to use searchlight analyses to probe for hue and luminance-polarity representations.

In summary, the present work used multivariate analyses of MEG data to quantitatively uncover the independent representations of hue and luminance polarity and their timing, which complements our previous analyses of the combined, interacting, representations of hue and luminance polarity that underlie the geometry of the neural representation of color[26]. The work reveals complex temporal dynamics that mediate the representation of color, which presumably reflect the involvement of color in many visual computations.

## Methods

**Visual stimuli**. Stimuli were eight square-wave spiral gratings on a neutral gray background (Fig. 1a)[59–61]. The eight stimulus colors, four hues at two luminance-contrast levels, of matched absolute cone contrast, were defined in DKL color space[17,56,57] using implementations by Westland and Brainard: the axes of this color space are defined in terms of activation of the two cone-opponent post-receptoral chromatic mechanisms (Supplementary Fig. 1). The $z$-axis is defined by luminance contrast. The four hues were defined by the intermediate axes of DKL space: at 45° (pink), 135° (blue), 225° (green), and 315° (orange). The absolute L-M cone modulation of the stimuli were matched; and the absolute S cone modulation of the stimuli were matched. Different colors were created by pairing different signs of modulation along L-M and S. Two spirals—one positive luminance contrast (+20° elevation; "light") and one negative luminance contrast (−20° elevation; "dark")—were created at each hue. The neutral adapting background was 33.5 cd/m². The luminance contrast of the stimuli was 25%, computed as Weber contrast because the stimuli were brief and not full field. Modulation of the cone-opponent mechanisms, shown in Supplementary Fig. 1, was computed relative to the adapting background gray. Using MATLAB scripts of Westland et al., the xyY values of the stimuli were converted to LMS values using xy2MB, and the LMS values were converted to DKL values using lms2dkl.

We designed the experiment such that (i) the stimuli were well above detection threshold, and (ii) the absolute cone contrasts of different hues (e.g., light pink versus light green) were comparable to the absolute cone contrasts of different luminance contrast levels (e.g., light pink versus dark pink). We can estimate the contrast of the stimuli in units of detection threshold, using detection data provided by Sachtler and Zaidi[17]. It is important to do this separately for color and luminance contrast because, as Sachtler and Zaidi showed, detection thresholds (measured in absolute cone contrast) are over 9× higher for luminance contrast than for color. The cone contrast is computed as the absolute value of the contrast in L, plus the absolute value of the contrast in M. For example:

We compute the luminance contrast carried by the pink hue as follows:

$$|(\text{Lcone}_{\text{light\_pink}} - \text{Lcone}_{\text{dark\_pink}})/(\text{Lcone}_{\text{light\_pink}} + \text{Lcone}_{\text{dark\_pink}})|$$
$$+ |(\text{Mcone}_{\text{light\_pink}} - \text{Mcone}_{\text{dark\_pink}})/(\text{Mcone}_{\text{light\_pink}} + \text{Mcone}_{\text{dark\_pink}})|$$
$$= |(27.6333 - 16.5006)/(27.6333 + 16.5006)| \quad (1)$$
$$+ |(14.2617 - 8.1970)/(14.2617 + 8.1970)| = 0.5225$$

The detection threshold ($|L|+|M|$) for luminance contrast is 0.014 (from Sachtler and Zaidi[17]). Thus the contrast of the stimuli was (0.5225/0.014), ~37× detection threshold.

We can compute the color contrast between pink and green at a given luminance contrast as follows:

$$|(\text{Lcone}_{\text{light\_pink}} - \text{Lcone}_{\text{light\_green}})/(\text{Lcone}_{\text{light\_pink}} + \text{Lcone}_{\text{light\_green}})|$$
$$+ |(\text{Mcone}_{\text{light\_pink}} - \text{Mcone}_{\text{light\_green}})/(\text{Mcone}_{\text{light\_pink}} + \text{Mcone}_{\text{light\_green}})| = 0.0525 \quad (2)$$

The detection threshold ($|L|+|M|$) for color contrast is 0.001724 (from Sachtler and Zaidi[17]). Thus the contrast of the light pink and green stimuli we used is (0.0525/0.001724), ~30× detection threshold. Comparable calculations for the dark pink and dark green stimuli show these stimuli to be ~51× detection threshold. The two values of color contrast (for light stimuli and dark stimuli) straddle the absolute contrast (in detection units) across luminance (37×, as calculated above). The possible difference in color contrast of light versus dark stimuli had no impact on the time course for decoding hue (see Supplementary Fig. 4), showing that at the supra-threshold values used, decoding strength is not likely influenced by stimulus saturation.

**MEGe acquisition and preprocessing**. Participants were scanned in the Athinoula A. Martinos Imaging Center of the McGovern Institute for Brain Research at the Massachusetts Institute of Technology (MIT) over the course of 2 sessions, on an Elekta Triux system (306-channel probe unit consisting of 102 sensor triplets, with 204 planar gradiometer sensors, and 102 magnetometer sensors). The experimental paradigm was created using Psychtoolbox[118]; stimuli were back-projected onto a 44" screen using a SXGA + 10000 Panasonic DLP Projector, Model No. PT-D10000U (50/60 Hz, 120 V). Data was recorded at a sampling rate of 1000 Hz, filtered between 0.03–330 Hz. Head location was recorded by means of five head position indicator (HPI) coils placed across the forehead and behind the ears. Before the MEG experiment began, three anatomical landmarks (bilateral pre-auricular points and the nasion) were registered with respect to the HPI coils, using a 3D digitizer (Fastrak, Polhemus, Colchester, Vermont, USA). During recording, pupil diameter and eye position data were collected simultaneously using an Eyelink 1000 Plus eye tracker (SR Research, Ontario, Canada) with fiber optic camera.

Once collected, raw data was preprocessed to offset head movements and reduce noise by means of spatiotemporal filters[119,120], with Maxfilter software (Elekta, Stockholm). Default parameters were used: harmonic expansion origin in head frame = [0 0 40] mm; expansion limit for internal multipole base = 8; expansion limit for external multipole base = 3; bad channels omitted from harmonic expansions = 7 s.d. above average; temporal correlation limit = 0.98; buffer length = 10 s. In this process, a spatial filter was applied to separate the signal data from noise sources occurring outside the helmet, then a temporal filter was applied to exclude any signal data highly correlated with noise data over time. Following this, Brainstorm software[121] was used to extract the peri-stimulus MEG data for each trial (−200 to 600 ms around stimulus onset) and to remove the baseline mean.

**Participants and task**. All participants ($N = 18$, 11 female, age 19–37 years) had normal or corrected-to-normal vision, were right-handed, spoke English as a first language, and provided informed consent. Participants received financial compensation ($30/h). One participant was an author and thus not naïve to the purpose of the study and was not paid for participating. During participants' first session, they were screened for colorblindness using Ishihara plates; they also completed a color-naming task as part of a separate study. After this task, participants completed a 100-trial practice session of the 1-back hue-matching task that would be used in the MEG experimental sessions. Once this was complete, participants were asked if they had any questions about the task or the experiment; eye-tracking calibration was performed; and MEG data collection began.

During the MEG data collection, participants were instructed to fixate at the center of the screen. Spirals were presented subtending 10° of visual angle for 116 ms, centered on the fixation point. The fixation point was a white circle that appeared during inter-trial intervals (ITIs, 1 s). In addition to the spirals, the words "green" and "blue" were presented in white on the screen for the same duration, and probe trials were presented with a white "?". (Responses to the words were analyzed as part of a separate study.) During the probe trials, which occurred every 3–5 stimulus trials (pseudo-randomly interspersed, 24 per run), participants were instructed to report via button press if the two preceding spirals did or did not match according to hue (1-back hue task). Maximum response time was 1.8 s, but the trials advanced as soon as participants answered.

Participants were encouraged to blink only during probe trials, as blinking generates large electrical artifacts picked up by the MEG. Each run comprised 100 stimulus presentations, and participants completed 25 runs per session over the course of approximately 1.5 h. Between each run, participants were given a break to rest their eyes and speak with the researcher if necessary. Once 10 s had elapsed, participants chose freely when to end their break by button-press. Over the course of both sessions, participants viewed each stimulus 500 times. Individual runs were identical across subjects, but the order of runs was randomized between subjects. The sequence of stimuli within each run was random with the constraint that the total number of presentations was the same for each stimulus condition over the set of runs obtained for each participant.

In the main experiments, data from all participants was used (no data was excluded because of poor behavioral performance).

All experimental procedures involving human participants, including the main experiment and all pilot and control experiments, were approved by the Wellesley College Institutional Review Board, the Massachusetts Institute of Technology Committee on the Use of Humans as Experimental Subjects, and the National Institutes of Health Intramural Institute Clinical Research Review Committee. Participants involved in all experiments, including the main experiment and all pilot and control experiments, provided informed consent to participate and to having their data published.

**Pilot experiment to determine number of trials per condition**. Prior to conducting the main experiments, we collected data in four participants (three females), using four colored stimuli (two hues, blue and orange, at light and dark contrast levels). Participants were instructed to fixate a small spot at the center of the display; besides this passive fixation, there was no task. Eye movements were monitored to ensure passive fixation. The goal of this pilot was to determine the number of trials needed to successfully decode color. As in the main experiment, the stimuli were spirals, subjects viewed each color 500 times, and each stimulus appeared for 100 ms followed by a 1 s ISI. Other details of experimental paradigm were the same as for the main experiment, except that the stimuli included only two hues at two luminance polarity values (four conditions total). The two hues were MB-DKL angles 150 and 300, which are intermediate colors corresponding roughly to blue and yellow; the luminance of the stimuli were: background gray, 41 cd/m²; positive luminance-contrast stimuli, 48–50 cd/m²; and negative luminance-contrast stimuli, 30–32 cd/m². To evaluate the experimental power, we set out to present each stimulus with what we thought was an excessive number of trials: 500 (leaving 375 after rejecting trials with eye blinks or other artifacts). Supplementary Fig. 2 shows the data reliability for the pilot experiment. The figure was generated by subsampling independent pairs of N% of data, computing the decoding for each independent set of data, and computing the correlation coefficient between the sets of data at each time point in the decoding curve. As expected, the test-retest curves plateau; but the plateau occurs only with substantial numbers of trials. Informed by these pilot experiments, we performed the main experiment with 500 trials. Preliminary results of these experiments have been presented and provide to our knowledge the first evidence for decoding color from MEG data[80,81]. The four participants who took part in this pilot experiment did not participate in the main experiments or the control experiment.

**Control experiment to measure impact of task**. Before launching the main experiments, we deployed a control experiment to determine the decoding parameters for the main experiments and to evaluate the task effects on decoding. This experiment was conducted in two participants (one female, age 20–30 years) who completed five sessions of 20 runs each, one of which had a technical glitch and was discarded. Each run had 100 stimulus presentations. Every 3–5 stimulus presentations, participants saw a "?" that prompted a response regarding the preceding two stimuli; the correct answer depended on the task for the run. For one half of the runs, the participants performed a 1-back hue matching task, and during the other half of the runs they performed a 1-back luminance-polarity matching task (Fig. 5a). Participants performed the same task for trials in a run, with the task alternating between runs for a given session. The results from the control experiment showed no impact of task performance on the latency or time-to-peak of decoding (Fig. 5b, c). For consistency in the main experiment, we had participants perform the same task. The decoding analysis shown in Fig. 5b, c, is for the identity problems illustrated in Fig. 1. The two participants who took part in the control experiment did not participate in the main experiments or the other pilot experiment. Exact $p$ values are reported throughout, except in for tests based on bootstrapping, where the number of bootstrap comparisons is zero. In those cases, the $p$ value can only be said to be less than 0.001 because there were 1000 bootstraps.

**MEG data processing and analysis**. Brainstorm software was used to process MEG data. Trials were discarded if they contained eyeblink artifacts, or contained out-of-range activity in any of the sensors (0.1–8000 fT). Three participants exhibited sensor activity consistently out of range, so this metric was not applied to their data as it was not a good marker of abnormal trials. After excluding bad trials, there were at least 375 good trials for every stimulus type for every participant.

Data were subsampled as needed to ensure the same number of trials per condition were used in the analysis.

Decoding was performed using the Neural Decoding Toolbox (NDT)[58]. We used the maximum correlation coefficient classifier in the NDT to train classifiers to associate patterns of MEG activity across the sensors with the visual stimuli presented. This classifier computes the mean population vector for sets of trials belonging to each class in the training data and calculates the Pearson's correlation coefficient between those vectors and the test vectors. The class with the highest correlation is the classifier's prediction. The main conclusions were replicated when using linear support vector machine classifiers. The classifiers were tested using held-out data—i.e., data that was not used in training. Data from both magnetometers and gradiometers were used in the analysis, and data for each sensor was averaged into 5 ms non-overlapping bins from 200 ms before stimulus onset to 600 ms after stimulus onset.

Custom MATLAB code was used to format MEG data preprocessed in Brainstorm for use in the NDT and to combine the two data-collection sessions for each participant. Decoding was performed independently for each participant, and at each time point. As illustrated in Fig. 2, for each decoding problem, at each timepoint (a 5 ms time bin), the 375 trials for each stimulus condition were divided into five sets of 75 trials. Within each set, the 75 trials were averaged together. This process generated five cross-validation splits: the classifier was trained on four of these sets, and tested on one of them, and the procedure was repeated five times so that each set was the test set once. This entire procedure was repeated 50 times, and decoding accuracies reported are the average accuracies across these 50 decoding "runs". This procedure ensured that the same data was never used for both training and testing, and it also ensured the same number of trials was used for every decoding problem. The details of the cross-validation procedure, such as the number of cross-validation splits, were determined during the pilot experiments to be those that yielded a high signal-to-noise ratio (SNR) and high decoding accuracy in both participants on the stimulus identity problem.

On each run, both the training and test data were z-scored using the mean and standard deviation over all time of the training data. Following others, we adopted a de-noising method that involved selecting for analysis data from the most informative sensors[63]; we chose the 25 sensors in the training data whose activity co-varied most significantly with the training labels. These sensors were identified as those with the lowest p-values from an F-test generated through an analysis of variance (ANOVA); the same sensors were then used for both training and testing. The sensor selection was specific for each participant. The sensors chosen tended to be at the back of the head (Fig. 8f, g). Analyses using all channels, rather than selecting only 25, yielded similar results.

All classification problems were binary (see Figs. 1 and 2). For each problem, a classifier was trained and tested in 5 ms bins from time $t = 200$ ms before stimulus onset to $t = 600$ ms after stimulus onset (Fig. 2). The classifiers' performance shown in Figs. 3 and 4 were generated through a bootstrapping procedure. First, the problems were evaluated for each participant, resulting in 18 independent decoding time courses for each unique problem. The decoding time courses for each problem were sampled 18 times with replacement and averaged, and this procedure was repeated 1000 times to produce 1000 time courses, which were averaged to generate the decoding traces in Figs. 3 and 4.

The 95% CI around the time to peak was determined from the times to peak for each of the 1000 bootstrapped time courses. Statistical tests on the difference in time to peak between two problems were performed using the bootstrap distributions of the differences in time to peak values. If the 95% confidence interval did not include 0, we rejected the null hypothesis, and p values were calculated based on the proportion of values that did fall below 0. To determine at which time points decoding accuracy was significantly above chance, a permutation test was used to calculate p values[122]. This was done by permuting the sign of the decoding accuracy data on a participant basis 1000 times. For each permutation sample, the mean accuracy was recomputed, resulting in an empirical distribution of 1000 mean accuracies. This distribution was used to convert the real mean accuracies across subjects over time to p-value maps over time. The significance threshold was $p < 0.05$, and significant regions were determined using a cluster-based approach (cluster defining threshold $p < 0.01$)[98,123]. Onset of significance was calculated as the first time point where accuracy was significant for four continuous 5 ms time bins—the requirement that the accuracy be significant for four consecutive bins was adopted to minimize false positives. Reported p values for paired comparisons of the timing and magnitude of the decoding accuracy are uncorrected.

For the control experiment, classifiers were trained and tested for each session individually to yield eight decoding time courses; the five cross validation splits consisted of 15 trials. The 95% confidence intervals on the times to peak and the statistical tests used to compare the peak times were produced similarly to those of the main experiment, by sampling from the eight decoding curves to produce 1000 bootstrapped time courses (shading in Fig. 5 shows the 95% CI at each time point). To estimate the significance of decoding at each time point, shown as the horizontal line of data points above the x-axis in the curves in Fig. 5, we used a within-subjects approach instead of the across-subjects approach used in the main experiment. A null distribution was produced to test whether the decoding at each time point was greater than what one would expect by chance. In each null decoding run, the stimuli identities were shuffled, and the classifier was trained and tested on the shuffled data. There were 1000 null decoding runs, and p values at

each time point were calculated based on the proportion of null decoding accuracies that exceeded the real decoding accuracy. These p values were then FDR corrected.

The test–retest curves to evaluate experimental power (Supplementary Figs. 2 and 3) were obtained by drawing pairs of independent samples of 10, 25, 40, and 50% of the trials (from a total of 375 trials), determining the correlation of the classification performance among the subproblems between the pairs, and repeating the procedure five times to generate error bars. For example, for the "10%" data point in the graph, we drew two sets of 10% of the trials at random—no trials were common to both sets. We trained separate classifiers on each of the independent sets and computed the temporal correlation between the two decoding time courses. We repeated this procedure 5×. We then averaged the correlation coefficients from the five repeats to obtain error bars.

In Fig. 6, we tested the performance of the classifiers across time: each classifier trained using data obtained at each time bin was tested using data obtained at every 5 ms time bin from −200 to 600 ms after stimulus onset creating a 2-dimensional matrix of decoding results. Significant time points were determined using a sign-permutation test and cluster correction (cluster-defining threshold $p < 0.01$)[98,123]. In Fig. 6c to compare the decoding of hue and luminance, the sign-permutation was carried out on a subproblem basis.

In Fig. 7, the 95% CI of the classification performance for each problem (the entries in Fig. 7b, c) were generated by bootstrapping across participants ($n = 1000$).

Note that decoding using pupil diameter and eye position cannot account for decoding of color, as discussed in a previous analysis of the MEGco data set[26].

**MRI dynamic localizer task**. To localize shape, place, face, and color-biased regions of interest (ROIs), 14 of 18 participants were scanned using the fMRI dynamic Localizer (DyLoc) described in Lafer Sousa et al., with the same parameters described there. In brief, participants passively viewed full color and grayscale (achromatic) versions of natural video clips that depicted faces, bodies, scenes, objects, and scrambled objects. Scrambled objects clips were clips in the object category that were divided into a 15 by 15 grid covering the frame, the boxes of which were then scrambled. Participants completed eight runs of the task, each of which contained 25 blocks of 18 s (20 stimuli and five gray fixation blocks). The stimuli were a maximum of 20° of visual angle wide and 15° tall. A Siemens 3 T MAGNETOM Prisma fit scanner (Siemans AG, Healthcare, Erlangen, Germany) with 64 RF receivers in the head coil was used to collect MRI data in 8 of 14 participants, while a Siemens 3 T MAGENTOM Tim Trio scanner with 32 channels in the head coil was used for the other six subjects.

For both groups, following Lafer-Sousa et al. a T2*-weighted echo planar imaging (EPI) pulse sequence was used to detect blood-oxygen-level-dependent (BOLD) contrast. Field maps (2 mm isotropic, 25 slices) were collected before each DyLoc run for the purpose of minimizing spatial distortions due to magnetic inhomogeneities in the functional volumes during analysis. Functional volumes (2 mm isotropic, 25 slices, field of view [FOV] = 192 mm, matrix = 96 × 96 mm, 2.0 s TR, 30 ms TE, 90° flip angle, 6/8 echo fraction) were collected on a localized section of the brain, aligned roughly parallel to the temporal lobe. The volumes covered V1–V4 in occipital cortex as well as the entirety of the temporal lobe ventral to the superior temporal sulcus (STS), and in some cases including parts of the STS. To allow for T1 equilibration, in each run, the first five volumes were not used during analysis.

High-resolution T1-weighted anatomical images were also collected for each subject by means of a multiecho MPRAGE pulse sequence (1 mm isotropic voxels, FOV = 256 mm, matrix = 256 × 256 mm).

**MRI analysis**. MRI data were processed following Lafer-Sousa et al (2016). Using Freesurfer (https://surfer.nmr.mgh.harvard.edu) and custom MATLAB scripts, the anatomical volumes were segmented into white-matter and gray-matter structures[124–126]. Functional data, processed on an individual subject basis, were field- and motion-corrected (by means of rigid-body transformations to the middle of each run), normalized for intensity after masking non-brain tissue, and spatially smoothed with an isotropic Gaussian kernel (3 mm FWHM) for better SNR. Subsequently, Freesurfer's bbregister was used to generate a rigid-body transformation used to align the functional data to the anatomical volume.

Whole-volume general linear model-based analyses were performed for all eight runs collected for each participant, using boxcar functions convolved with a gamma hemodynamic response function as regressors[127]; each condition's boxcar function included all blocks from that condition, as well as nuisance regressors for motion (three translations, three rotations) and a linear trend to capture slow drifts.

Brain regions used to restrict decoding analyses of MEG source data were defined using two methods. Anatomically defined regions were defined using surface-based Freesurfer atlases: "precentral" and "frontal" regions corresponded respectively to the "precentral" and "rostralmiddlefrontal" bilateral regions in the Desikan–Killiany atlas[77]; V1 (BA 17), V2 (BA 18), and MT regions were defined using the Brodmann atlas (https://surfer.nmr.mgh.harvard.edu/fswiki/BrodmannAreaMaps; Brodmann, 1909). Functionally defined regions were defined individually using Lafer-Sousa et al.[76] as a reference. FFA was selected from voxels where face response > object response ($p < 0.001$), using data from all eight runs.

The same procedure was followed for VVP-c from voxels where color response > grayscale response, PPA from voxels where scene response > object response, and LO from voxels where object response > scrambled object response.

**Source localization and decoding within fMRI regions of interest**. Current source density is a metric representing the current at each point on the surface of the brain, defined by the source grid. First, using Brainstorm, a minimum norm estimate (MNE) was calculated, which was "depth-weighted", to compensate for a bias in current density calculations that results in more activity being placed on superficial gyrii, neglecting regions of cortex embedded in deeper sulci. The MNE at a given source was normalized by the square root of a local estimate of noise variance (dynamical Statistical Parametric Mapping; dSPM)[128], yielding a unitless z-scored statistical map of activity. Once a source map was created, ROI analysis was performed by restricting the features of the classifiers to the top 25 sources within the bounds of a given ROI whose activity covaried most with the training labels, using custom code. Additionally, the sources within an ROI were averaged together within subjects to yield the average sensor response by ROI.

**Reporting summary**. Further information on research design is available in the Nature Research Reporting Summary linked to this article.

## Data availability

Source data are provided with the paper. The MEGco dataset[26] is published and can be accessed at the OpenNeuro data base[129] (https://doi.org/10.18112/openneuro.ds003352.v1.0.0) and at NEICOMMONS[130] (https://neicommons.nei.nih.gov/#/MEGco). Source data are provided with this paper.

## Code availability

The procedures and code used to produce all the figures and statistical analyses are available at NEICOMMONS[131] (https://neicommons.nei.nih.gov/#/TempDynamicsHueLum).

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

## Acknowledgements

We thank Chris Baker, Susan Wardle, Santani Teng, and Qasim Zaidi for helpful discussions, and Stuart Duffield for help checking and documenting the analysis pipeline, and Mikhail Laryukhin and the NEICOMMONS team for help making the data publically available. This research was supported in part by the Intramural Research Program of the National Institutes of Health, National Eye Institute ZIAEY000558 (B.R.C.).

## Author contributions

K.H. and B.R.C. conceived and designed the experiments; K.H. and I.R. collected the data; K.H., I.R., and S.S. analyzed the data; I.R., S.S., and B.R.C. made the figures; D.P. provided expertise and resources for MEG; B.R.C. supervised the work and wrote the paper.

## Funding

## Competing interests

The authors declare no competing interests.
