## [Peer Review File · Nature Communications]

Temporal dynamics of the neural representation of hue and luminance polarityEditorial Note: This manuscript has been previously reviewed at another journal that is not operating a transparent peer review scheme. This document only contains reviewer comments and rebuttal letters for versions considered at *Nature Communications*. Mentions of prior referee reports have been redacted.

REVIEWER COMMENTS

Reviewer #2 (Remarks to the Author):

I reviewed a previous version of this manuscript [REDACTED] and I read the current manuscript in light of my previous comments and the author rebuttal to those comments. My overall assessment of the manuscript as interesting basic research holds, and I would like to see this work eventually published. I think my previous concerns have been adequately addressed by the revised manuscript. I have a few remaining comments:

1. Typo in Abstract: subtlety → subtly?
2. Results, middle of first paragraph: would be good to specify the units for the modulation magnitude for hue and lum contrast here, as right now they are only explained at the end of the results section and in methods.
3. Results, page 21: Did I understand correctly that hue was only decodable in V2 and not in color-selective voxels identified by the functional localizer? I appreciate that the authors mention the trickiness of MEG source localization in the discussion, because although I like this analysis, it is not clear how accurately the authors can actually select the channels to analyze based on the source localization together with the functional anatomy.
4. Are the reported p-values in paired comparisons throughout the manuscript uncorrected? I could not find mention of this in the results or methods.

Reviewer #3 (Remarks to the Author):

This is a re-analysis of the recent Rosenthal et al. MEG dataset, using the same decoding approach to investigate how luminance contrast and hue are processed in the cortex. This is a very nice approach and is the next logical step for what to do with this rich dataset, following on from the analyses presented in the authors' previous paper.

There is however one major issue with the manuscript. The authors claim that they are decoding luminance contrast but their stimuli differ in luminance polarity rather than luminance contrast. Here, the stimuli have the Weber contrast of 26%, either in a light or dark polarity. So in effect, the authors cannot decode luminance contrast – its absolute value is the same for all 8 patches, at 26%. What they can try to decode is the sign i.e. polarity of this contrast. Reviewer 2 also commented on this. To evaluate decoding of hue or luminance contrast, as they state, the authors would need stimuli with at least two variations of contrast of the same polarity, as done in the study by Sutterer et al. (2021, *Psychophysiology*). I understand that the current manuscript was probably submitted and in review prior to the publication of Sutterer et al., so I am only referring to this paper as an example of a dataset that allows decoding of luminance contrast. In light of this, it seems that the manuscript would need to undergo a major change of terminology to accurately reflect what the authors have actually done here - changing all instances of 'luminance contrast' to 'luminance polarity' and restating some of the conclusions to be more in the line with what the authors did.

I also disagree with the interpretation that “patterns of activity in the brain associated with hue are more stable” (p.16) – just because the luminance-driven signal is more transient, does not mean that

during that short period of time it is not stable. And indeed, VEPs have very high global field power during luminance-driven early responses. I suggest changing “more and less stable” dichotomy to “transient” vs. “sustained”, as lack of generalisation across time does not imply signal stability, but merely its transience.

Page 3 “But it has not been possible to infer from these experiments the underlying neural mechanisms because all subcortical channels respond to equiluminant stimuli”. The study by Logothetis et al. (1990) does show that this is the case in macaques viewing motion, texture and depth stimuli at nominal equiluminance. At observer equiluminance, this would not be the case, as discussed in the authoritative review by Gegenfurtner and Kiper (2003): “The existence of residual responses at photometric isoluminance (the 0 point on the abscissa of Figure 8) in cortical areas that do not contain many color-opponent neurons has important implications for the interpretation of studies that use isoluminant stimuli to isolate the cortical mechanisms involved in color perception (see Logothetis et al. 1990). If such studies use stimuli that are at photometric isoluminance, they will evoke responses in areas that are expected to remain silent.” I suggest adding the word nominally, i.e. “because all subcortical channels respond to nominally equiluminant stimuli”.

Despite comments from Reviewer 1, the authors’ did not change the following statement in the introduction: “Moreover, such experiments are inconclusive about timing because response latency depends on stimulus contrast, and there is no consensus on how to equate color contrast and luminance contrast”. There are in fact quite a few valid approaches for equating contrast, but the context of this sentence seems to imply that there is no good solution (something that is reiterated again on p.8), rather than that there are a couple of good enough solutions, whose usefulness may depend on what the contrast needs to be equated for (e.g. detection, discriminability or perceived contrast). A recently published paper by Hardman & Martinovic, 2021 summarises some of these methods, but it was Bloj, Finlayson and colleagues who led on suprathreshold contrast matching between colour and luminance much earlier, while trying to establish mappings between color and luminance contrast for the purposes of printing. Thus, these sections should be rewritten to present a more balanced view of the literature.

Finally, the sentence in the conclusion overstates the study’s findings on p23: “But the present analysis provide to our knowledge the first direct neural evidence that the human brain has independent representations for hue and luminance contrast and the timing of these representations.” In fact, the existing body of EEG work, from 1980s onwards (Berninger et al., 1989, Rabin et al., 1994, multiple papers from Murray, Kulikowski and colleagues, multiple papers from Crognale and colleagues) had demonstrated that the early onset P1 component is specific for luminance, and that the peak of the subsequent N1 peak is delayed by ~20 ms when driven by chromatic information alone (this negative peak is labelled as cVEP in this literature). Different maps of P1 on the isoluminant N1 are already evidence of different neural sources. Similarly, previous macaque work already showed that while most neurons respond to both color and luminance, some do respond only to 1 type of contrast (as covered in many reviews). Thus this sentence is a vast overstatement of the study being “first direct neural evidence”.

Reviewer #4 (Remarks to the Author):

The manuscript presents an MEG decoding study of hue and luminance contrast. The results show both these features are independently decodable from MEG, with different temporal dynamics.

I was asked to specifically focus on the MEG decoding methods, and in my opinion these are of the highest standard. I thus have no comments regarding the methodology.

I have one minor suggestion regarding the task-based results (Figure 4). The findings showed longer decoding duration when the stimulus is task relevant. However, the 1-back task has a memory component, so perhaps some speculation can be included discussing the extent to which the task-

effects reflect memory processes. See e.g., recent work from our group using an n-back task and associated changes in decoding duration reflecting memory requirements:
<https://doi.org/10.51628/001c.21174>

Response to reviews

Below is our point-by-point response to reviews. Reviewer comments are in black; our responses are in blue.

Reviewer #2 (Remarks to the Author):

I reviewed a previous version of this manuscript [REDACTED] and I read the current manuscript in light of my previous comments and the author rebuttal to those comments. My overall assessment of the manuscript as interesting basic research holds, and I would like to see this work eventually published. I think my previous concerns have been adequately addressed by the revised manuscript. I have a few remaining comments:

Thank you for the constructive feedback on the prior submission [REDACTED] and this round. We hope we have addressed all your concerns, and that the resulting paper makes a stronger, clearer contribution.

I have a few remaining comments:

1. Typo in Abstract: subtlety → subtly?

Fixed, thanks.

2. Results, middle of first paragraph: would be good to specify the units for the modulation magnitude for hue and lum contrast here, as right now they are only explained at the end of the results section and in methods.

We have moved the description of the units for the modulation for hue and luminance contrast to the first part of the results. We have also revised the description of the stimuli to recognize a comment by another reviewer who pointed out that the stimuli do not vary parametrically in luminance contrast, but rather in sign of luminance contrast.

3. Results, page 21: Did I understand correctly that hue was only decodable in V2 and not in color-selective voxels identified by the functional localizer? I appreciate that the authors mention the trickiness of MEG source localization in the discussion, because although I like this analysis, it is not clear how accurately the authors can actually select the channels to analyze based on the source localization together with the functional anatomy.

Decoding hue generalized across the sign of luminance contrast was not reliable in any ROI. There is a hint of possible decoding in V1 and/or V2, which varies if we re-run the bootstrapping, as we discovered in generating the source-data file for the raw data needed to generate the display items. We have revised the text accordingly. It is not clear to use why the analysis fails to recover hue decoding in color-biased regions defined with the functional localizer. One possibility is that the source localization algorithm is inaccurate, which is not inconceivable given there is no unique solution for source localization. Another, not mutually exclusive, possibility is that the number of voxels included in the analysis is too few to enable significant decoding.

4. Are the reported p-values in paired comparisons throughout the manuscript uncorrected? I could not find mention of this in the results or methods.

The tests for determining time points of significant decoding are FDR corrected; the p values for paired comparisons, e.g. of the time-to-peak and the magnitude of decoding, are uncorrected, which is now stated in the methods.

Reviewer #3 (Remarks to the Author):

This is a re-analysis of the recent Rosenthal et al. MEG dataset, using the same decoding approach to investigate how luminance contrast and hue are processed in the cortex. This is a very nice approach and is the next logical step for what to do with this rich dataset, following on from the analyses presented in the authors' previous paper.

Thanks for the helpful feedback. We hope that we have adequately addressed the points you raise.

There is however one major issue with the manuscript. The authors claim that they are decoding luminance contrast but their stimuli differ in luminance polarity rather than luminance contrast. Here, the stimuli have the Weber contrast of 26%, either in a light or dark polarity. So in effect, the authors cannot decode luminance contrast – its absolute value is the same for all 8 patches, at 26%. What they can try to decode is the sign i.e. polarity of this contrast. Reviewer 2 also commented on this. To evaluate decoding of hue or luminance contrast, as they state, the authors would need stimuli with at least two variations of contrast of the same polarity, as done in the study by Sutterer et al. (2021, Psychophysiology). I understand that the current manuscript was probably submitted and in review prior to the publication of Sutterer et al., so I am only referring to this paper as an example of a dataset that allows decoding of luminance contrast. In light of this, it seems that the manuscript would need to undergo a major change of terminology to accurately reflect what the authors have actually done here - changing all instances of 'luminance contrast' to 'luminance polarity' and restating some of the conclusions to be more in the line with what the authors did.

We have made it explicit in the abstract and results that the experiments afford decoding of the sign of luminance contrast, not luminance contrast. We had taken the original comment by Reviewer 2 to be a critique of the use of the term “lightness” or “grayscale equivalent”. We now see the potential confusion regarding the term “luminance contrast” and have changed most of these instances to “luminance polarity”. We have held onto the term “luminance contrast” when it seems more appropriate. We have also included a reference to the recently published Sutterer paper.

I also disagree with the interpretation that “patterns of activity in the brain associated with hue are more stable” (p.16) – just because the luminance-driven signal is more transient, does not mean that during that short period of time it is not stable. And indeed, VEPs have very high global field power during luminance-driven early responses. I suggest changing “more and less stable” dichotomy to “transient” vs. “sustained”, as lack of generalisation across time does not imply signal stability, but merely its transience.

We have changed the “more and less stable” dichotomy to “transient” vs. “sustained”. Thank you for this suggestion.

Page 3 “But it has not been possible to infer from these experiments the underlying neural mechanisms because all subcortical channels respond to equiluminant stimuli”. The study by Logothetis et al. (1990) does show that this is the case in macaques viewing motion, texture and depth stimuli at nominal equiluminance. At observer equiluminance, this would not be the case, as discussed in the authoritative review by Gegenfurtner and Kiper (2003): “The existence of residual responses at photometric isoluminance (the 0 point on the abscissa of Figure 8) in cortical areas that do not contain many color-opponent neurons has important implications for the interpretation of studies that use isoluminant stimuli to isolate the cortical mechanisms involved in color perception (see Logothetis et al. 1990). If

such studies use stimuli that are at photometric isoluminance, they will evoke responses in areas that are expected to remain silent.” I suggest adding the word nominally, i.e. “because all subcortical channels respond to nominally equiluminant stimuli”.

We are happy to add the word “nominally” (and have done so), but we may disagree with the implication in the quote from the G&K review, that there are cortical areas that do not respond to color. The difficulty in interpreting results using equiluminant stimuli versus luminance-contrast stimuli is in determining which metric to use to equate the two. Johnson and colleagues (2001, 2004, 2008) equated color stimuli and luminance-contrast stimuli for contrast and found that most neurons in V1 respond well to both stimuli. We do not think that Johnson’s results are not in conflict with the data cited by the G&K review—the two studies simply use different stimuli contrasts. The key difference is that the stimuli away from the 0 point on the abscissa in Figure 8 of the G&K review would have higher absolute cone contrast than the stimuli at the 0 point. In the Johnson framework, the difference in response magnitude across the abscissa reflects a difference in absolute cone contrast of the stimuli, and is not a clue to whether the neurons contribute to color processing or not. Also relevant to this discussion is the work from Hubel and Livingstone (1990), who show that the equiluminant null point varies widely across parvocellular neurons in the LGN. We think those results predict why most neurons in V1 respond to both luminance contrast and equiluminant stimuli, even if the stimuli are equated for cone contrast: there are not separate mosaics for L-ON and M-ON geniculate inputs to the cortex, and V1 cells will sample both mosaics as needed to optimize orientation tuning (Conway and Livingstone, 2006). These considerations are not intended to argue against the idea that there are areas that are more responsive to color, but simply to rebut the notion that it is well established that there are areas that are not responsive to color. Space constraints in the present manuscript preclude a fuller treatment of these issues.

Despite comments from Reviewer 1, the authors’ did not change the following statement in the introduction: “Moreover, such experiments are inconclusive about timing because response latency depends on stimulus contrast, and there is no consensus on how to equate color contrast and luminance contrast”. There are in fact quite a few valid approaches for equating contrast, but the context of this sentence seems to imply that there is no good solution (something that is reiterated again on p.8), rather than that there are a couple of good enough solutions, whose usefulness may depend on what the contrast needs to be equated for (e.g. detection, discriminability or perceived contrast). A recently published paper by Hardman & Martinovic, 2021 summarises some of these methods, but it was Bloj, Finlayson and colleagues who led on suprathreshold contrast matching between colour and luminance much earlier, while trying to establish mappings between color and luminance contrast for the purposes of printing. Thus, these sections should be rewritten to present a more balanced view of the literature.

We have edited the quoted sentence to say “Moreover, such experiments are inconclusive about timing because response latency depends on stimulus contrast, and there is no single method for equating color contrast and luminance contrast”. We reference the Shevell and Kingdom paper following this sentence because our sentence is almost a direct quote from their paper, where they say “A difficulty, however, arises in comparing chromaticity and luminance for a given form/motion task because performance usually improves with stimulus contrast, and there is no common metric to equate chromatic contrast and luminance contrast.” Shevell and Kingdom go on to describe one approach that aims to overcome the difficulty. But the difficulty nonetheless remains, as implied by Connah Finlayson and Bloj (2008),

who write that “in general, the newer techniques can provide a greyscale image which is preferred to that derived by luminance, especially for images that have prominent equiluminant boundaries. The results also show that this advantage is not guaranteed for every image, and that no particular algorithm provides consistently better performance than the others”. We recognize our original text might have been taken to imply that it is simply impossible to equate luminance and color (or that there are no good-enough solutions depending on the objective, e.g. to equate for detection, discrimination, or perceived contrast). We have revised the results with the hopes of providing a more balanced treatment of the literature, including references to a range of papers on the topic.

Finally, the sentence in the conclusion overstates the study’s findings on p23: “But the present analysis provide to our knowledge the first direct neural evidence that the human brain has independent representations for hue and luminance contrast and the timing of these representations.” In fact, the existing body of EEG work, from 1980s onwards (Berninger et al., 1989, Rabin et al., 1994, multiple papers from Murray, Kulikowski and colleagues, multiple papers from Crognale and colleagues) had demonstrated that the early onset P1 component is specific for luminance, and that the peak of the subsequent N1 peak is delayed by ~20 ms when driven by chromatic information alone (this negative peak is labelled as cVEP in this literature). Different maps of P1 an the isoluminant N1 are already evidence of different neural sources. Similarly, previous macaque work already showed that while most neurons respond to both color and luminance, some do respond only to 1 type of contrast (as covered in many reviews). Thus this sentence is a vast overstatement of the study being “first direct neural evidence”.

Thank you for directing us to these references. We have revised the discussion, removing the clause “to our knowledge the first”, and included a discussion of the references.

Reviewer #4 (Remarks to the Author):

The manuscript presents an MEG decoding study of hue and luminance contrast. The results show both these features are independently decodable from MEG, with different temporal dynamics.

I was asked to specifically focus on the MEG decoding methods, and in my opinion these are of the highest standard. I thus have no comments regarding the methodology.

Thanks!

I have one minor suggestion regarding the task-based results (Figure 4). The findings showed longer decoding duration when the stimulus is task relevant. However, the 1-back task has a memory component, so perhaps some speculation can be included discussing the extent to which the task-effects reflect memory processes. See e.g., recent work from our group using an n-back task and associated changes in decoding duration reflecting memory requirements:

<https://doi.org/10.51628/001c.21174>

We have included in the discussion a description of the possible impacts of attention and task on the time course of the decoding results, including a reference to this paper. Thanks for directing us to this literature.

REVIEWER COMMENTS

Reviewer #2 (Remarks to the Author):

The authors have addressed all of my comments.

Reviewer #3 (Remarks to the Author):

I would like to thank the authors for carefully considering and addressing all of my suggestions.

I have only spotted the following typo:

"We lack of an impact of task would have been predicted by prior work" should be "The lack of an impact..."

Reviewer #4 (Remarks to the Author):

No further comments

We thank the reviewers for their constructive feedback and comments. Point-by-point response, as requested by the journal, are provided below.

REVIEWERS' COMMENTS

Reviewer #2 (Remarks to the Author):

The authors have addressed all of my comments.

thank you.

Reviewer #3 (Remarks to the Author):

I would like to thank the authors for carefully considering and addressing all of my suggestions.

Thank you

I have only spotted the following typo:

"We lack of an impact of task would have been predicted by prior work" should be "The lack of an impact..."

Fixed.

Reviewer #4 (Remarks to the Author):

No further comments

Thank you

REVIEWER COMMENTS

Reviewer #3 (Remarks to the Author):

I was never fully convinced by the weak but significant effect that was interpreted as indicating that colour decoding in the presence of luminance contrast behaves in line with the daylight locus. It was always much more likely that EEG signal is sensitive to different behaviour of L-M and S-(L+M) signals in the presence of luminance contrast, as argued by the current revision and already demonstrated by Martinovic & Andersen (2018; Neuroimage) using steady-state visual evoked potentials. Shapley and colleagues also provide highly relevant work on this topic using visual evoked potentials, and previous work on the topic using EEG was done by Rabin, Crognale, Valberg and others. So in this sense, I find the current results more coherent with my understanding of colour-elicited VEPs than the previous version of results.

However, some parts of the manuscript will need revising because they make claims that are not fully in line with previous literature. For example the final sentence of the abstract now states that “luminance contrast is mediated by both L-M and S cone sub-cortical mechanisms.” As I pointed out in my previous comments, the authors only have a single value of luminance contrast in their study, so it is not logically coherent to say that luminance contrast is mediated by another factor when it is in fact constant. It might be better to say that L-M and S-cone signals combine differently with luminance contrast, which we already know from VEP and SSVEP literature, rather than argue that “luminance-polarity is encoded by a combination of both L-M and S chromatic mechanisms.” It is not likely at all that L-M and S mechanisms encode luminance polarity, but it is quite likely and consistent with literature to say that joint processing of L-M/S signals and luminance contrast differs. The introduction now says that “the extent to which luminance is carried by both cone-opponent retinal mechanisms (L-M and S) remains unclear”, when in fact much previous works addresses combined processing of colour and luminance signals. The authors should conduct an appropriate literature review and also be mindful of terminology they use, whether talking about luminance, luminance contrast, or luminance polarity. This also applies to the following claim: “If hue and luminance contrast are encoded separately, one might expect luminance contrast to be computed earlier than hue because magnocellular neurons have shorter latencies than parvocellular neurons. But because there are relatively fewer magnocellular neurons, their latency advantage may be lost through convergence in visual cortex. The time taken by the brain to encode hue and luminance contrast remains poorly understood.” This is again an overstatement, which ignores the well-known fact that above certain low spatial frequencies, it is parvocellular neurons that combine luminance and colour contrast. Colour and luminance signals overlap in their cortical processing, which has already been shown by visual evoked potential work of Shapley and colleagues (e.g. Xing et al., 2015, Journal of Neuroscience).

There is also a new section in the discussion that needs to be amended, starting with the following sentence: “The success of the classifiers might be surprising given EEG and MEG are thought to show coarse spatial resolution, and the spatial scale of color columns, in early visual cortex or in higher visual cortex is relatively fine.” This is an incorrect reduction of classification from EEG/MEG to a simply spatial problem. If signals have different temporal signatures, then the spatial maps will be displaced in time in their evolution, resulting in significantly different waveforms and hence above-chance decoding. The authors should remove this section or rephrase it to emphasise how temporal resolution of EEG/MEG also contributes to differentiation of signals.

The conclusion that “results provide partial support for the hypothesis” of daylight-locus is overstated. None of the results that evaluate this hypothesis were significant. Poorer decoding for blue is not consistent with this prediction, as it should have been both blue and orange. There are more parsimonious ways to explain blue being the odd colour, and the authors cite some of this literature. Thus, this is not partial support but rather means that this information is unlikely to be carried in the EEG signal, at least.

Reviewer #4 (Remarks to the Author):

I have read the revised documents from Hermann et al. and I have a few minor points that I think should be addressed:

(1) The authors subsample trials to equalise the peaks and make the signal onsets more comparable. While this is good practice, it still deserves a cautionary statement in the interpretation, as it only 'corrects' for the peak strength, and not other factors that could cause differences in decoding onsets, for example between trial temporal variability. Furthermore, the authors repeatedly attribute this procedure to Cichy et al., 2014, but this procedure was to the best of my knowledge not used or described in Cichy et al 2014. Perhaps the authors meant to cite Isik et al., 2014 instead who I believe were among the first to use this procedure in MEG decoding: Isik L, Meyers EM, Leibo JZ, Poggio T. The dynamics of invariant object recognition in the human visual system. *J Neurophysiol* 111, 91-102 (2014).

(2) Several figures contain only 2 ticks on the axes (e.g. xaxis with 0ms and 600ms, and a yaxis of 50% and 85%). Increasing the number of ticks on those axes would make the results easier to interpret.

(3) The authors need to carefully check the reference list. It contains several references to preprints that should be replaced with their published journal versions. I know that the preprints listed below have now been published, so they at least need updating, but I encourage the authors to do another pass before this paper is published. The list also contains a reference to the biorxiv version of the current manuscript.

- 88 Hermann K, Rosenthal I, Singh S, Pantazis D, Conway BR. Temporal dynamics of the neural mechanisms for encoding hue and luminance contrast uncovered by magnetoencephalography. *BioRxiv* <https://www.biorxiv.org/content/10.1101/2020.06.17.155713v2>, (2020).

- o this seems to be the current manuscript

- 63 Grootswagers TR, A. K.; Shatek, S. M.; Carlson, T. A. . The neural dynamics underlying prioritisation of task- relevant information. *arXiv:210201303v2* (2021).

- o published in *NBDT*

- 80 Hajonides JE, Nobre AC, van Ede F, Stokes MG. Decoding visual colour from scalp electroencephalography measurements. *BioRxiv* <https://doi.org/10.1101/2020.07.30.228437> (2020)

- o published in *neuroimage*

- 107 King J-R, Wyart V. The Human Brain encodes a Chronicle of Visual Events at each Instant of Time. *bioRxiv*, 846576 (2019).

- o published in *journal of neuroscience*

REVIEWER COMMENTS

Reviewer #3 (Remarks to the Author):

I was never fully convinced by the weak but significant effect that was interpreted as indicating that colour decoding in the presence of luminance contrast behaves in line with the daylight locus. It was always much more likely that EEG signal is sensitive to different behaviour of L-M and S-(L+M) signals in the presence of luminance contrast, as argued by the current revision and already demonstrated by Martinovic & Andersen (2018; Neuroimage) using steady-state visual evoked potentials. Shapley and colleagues also provide highly relevant work on this topic using visual evoked potentials, and previous work on the topic using EEG was done by Rabin, Crognale, Valberg and others. So in this sense, I find the current results more coherent with my understanding of colour-elicited VEPs than the previous version of results.

We thank the reviewer for reviewing the manuscript yet again. The feedback has improved the manuscript. We appreciate the pointers to these papers. We have now included citations for the Marinovic and Andersen paper, Bilmer et al, and Xing et al, regarding their work describing the interactions of color and luminance in perception, and we retain in the revision the citation to the important work by Rabin and others.

Two original contributions of the present work are the characterization of the timing differences that is afforded by the multivariate analysis, and the determination of the extent to which luminance contrast can be decoded from representations elicited by intermediate directions in color space. These topics—especially the second—has not been definitively addressed in the literature on visual evoked potentials.

In the stimulus design, all stimuli modulate both S and L-M dimensions. If luminance representations are determined by the contribution of the S-cone mechanism, then luminance-polarity representations using stimuli that modulate the S cone with the same sign, regardless of the L-M contribution, should be comparably decodable. Contrary to this prediction, the results show a striking difference in luminance decoding when the stimuli modulate +S+(M-L) versus +S+(L-M). This asymmetry is not consistent with a simple interpretation of the contribution of the S mechanism to luminance representations, but it is consistent with the idea that luminance representations are asymmetric with respect to the daylight locus. We do not find a similar asymmetry between -S+(M-L) versus -S+(L-M), which is why we had stated that the results provide “partial support” for the hypothesis. In other words, the results show that only one pole of the daylight axis is compromised in encoding luminance contrast, the pole associated with the +S+(M-L) axis. But the lack of asymmetry between -S+(M-L) versus -S+(L-M) does not actually undermine the positive evidence for the hypothesis, which strongly argues against the alternative possibility (that the interaction of color and luminance are solely accounted for by an interaction with the S-cone axis). We have clarified these arguments in the paper.

However, some parts of the manuscript will need revising because they make claims that are not fully in line with previous literature. For example the final sentence of the abstract now states that “luminance contrast is mediated by both L-M and S cone sub-cortical mechanisms.” As I pointed out in my previous comments, the authors only have a single value of luminance contrast in their study, so it is not logically coherent to say that luminance contrast is mediated by another factor when it is in fact constant. It might be better to say that L-M and S-cone signals combine differently with luminance contrast, which we already know from VEP and SSVEP literature, rather than argue that “luminance-polarity is encoded by a combination of both L-M and S chromatic mechanisms.” It is not likely at all that L-M and S mechanisms encode luminance polarity, but it is quite likely and consistent with literature to say that joint processing of L-M/S signals and luminance contrast differs.

The quoted passage omitted the key word “suggests” that was used in the original text to indicate that the conclusion is not definitively proven with the current data alone. But to avoid any confusion, we have revised the abstract to state that the pattern of results is consistent with observations that luminance contrast is mediated by both L-M and S cone sub-cortical mechanisms.

The introduction now says that “the extent to which luminance is carried by both cone-opponent retinal mechanisms (L-M and S) remains unclear”, when in fact much previous work addresses combined processing of colour and luminance signals. The authors should conduct an appropriate literature review and also be mindful of terminology they use, whether talking about luminance, luminance contrast, or luminance polarity.

We have cited additional work as described in the first response above. We believe that the introduction as written is consistent with the literature: there are papers that argue for a segregation of mechanisms encoding hue and luminance (contrast or polarity), and there are papers that imply or assert admixing of these mechanisms. This range of conclusions suggests that the extent to which luminance is carried by both cone-opponent mechanisms remains unclear.

In prior rounds of review, the reviewer raised the excellent point that “luminance contrast” is potentially misleading when used to describe a stimulus of changes in sign of luminance contrast without changes in magnitude of luminance contrast. We have previously rectified this potential confusion. Those remaining occasions when we discuss “luminance contrast” are in reference to what our results might suggest about how luminance contrast is encoded or represented more generally, rather than the restricted case of the sign of luminance contrast. We note that the luminance contrast in the paper included two levels, +1 and -1 (multiplied by a fixed magnitude), and we believe results using this stimulus design are sufficient to contribute to the understanding of how luminance contrast is encoded by the nervous system.

This also applies to the following claim: “If hue and luminance contrast are encoded separately, one might expect luminance contrast to be computed earlier than hue because magnocellular

neurons have shorter latencies than parvocellular neurons. But because there are relatively fewer magnocellular neurons, their latency advantage may be lost through convergence in visual cortex. The time taken by the brain to encode hue and luminance contrast remains poorly understood.” This is again an overstatement, which ignores the well-known fact that above certain low spatial frequencies, it is parvocellular neurons that combine luminance and colour contrast. Colour and luminance signals overlap in their cortical processing, which has already been shown by visual evoked potential work of Shapley and colleagues (e.g. Xing et al., 2015, Journal of Neuroscience).

We are well-aware of the role of parvocellular cells in encoding both luminance contrast and color, as evident in an earlier version of this manuscript, quoted here from the Biorxiv preprint:

2020.06.17.155713v1.full.pdf (biorxiv.org) “One prominent theory, succinctly articulated by Gegenfurtner (2003), is that hue is “processed not in isolation, but together with information about luminance [contrast] and visual form, by the same neural circuits, to achieve a unitary and robust representation of the visual world” (Gegenfurtner, 2003). This position is supported by neurophysiological recordings in the lateral geniculate nucleus (LGN) of monkeys. Parvocellular LGN cells multiplex tuning for luminance contrast and color opponency (Reid and Shapley, 2002, Wiesel and Hubel, 1966). Indeed, the response properties of parvocellular cells correspond to the relative spatial frequency of luminance-contrast vision (high-pass) and color vision (low-pass) (De Valois and Switkes, 1983, Granger and Heurtley, 1973, Mullen, 1985, van der Horst and Bouman, 1969). But besides parvocellular neurons, the LGN contains magnocellular cells, which respond to luminance contrast and are not cone opponent; and koniocellular neurons, some of which respond to S-cone signals and may represent a distinct chromatic channel (Lee et al. , 1989, Martin et al. , 1997). These observations animate an alternative idea, that the LGN encodes hue and luminance contrast in separable channels (Dobkins, 2000).”

The introduction from which this passage was quoted was revised to accommodate length constraints. The passage was deemed unnecessary because these facts are reasonably well known and, importantly, because they have no bearing on the argument that magnocellular neurons could contribute to luminance contrast (or luminance polarity) representations. As Maunsell and colleagues point out in the cited paper, if magnocellular neurons are responsible (or contribute) to the representation of luminance contrast (or luminance polarity), the representation of luminance contrast (or polarity) might be expected to arise earlier. If, as the reviewer maintains, it is established knowledge that luminance contrast (or polarity) and color are encoded jointly, then one would predict that luminance polarity and hue should be decoded with the same time course. Our results provide compelling evidence against this prediction, underscoring the originality of the results.

There is also a new section in the discussion that needs to be amended, starting with the following sentence: “The success of the classifiers might be surprising given EEG and MEG are thought to show coarse spatial resolution, and the spatial scale of color columns, in early visual cortex or in higher visual cortex is relatively fine.” This is an incorrect reduction of classification

from EEG/MEG to a simply spatial problem. If signals have different temporal signatures, then the spatial maps will be displaced in time in their evolution, resulting in significantly different waveforms and hence above-chance decoding. The authors should remove this section or rephrase it to emphasise how temporal resolution of EEG/MEG also contributes to differentiation of signals.

We respectfully stand by our statement for the following reason: A different temporal evolution in the response to two conditions will only be manifest in the decoding analysis if the two conditions are also accompanied by different spatial representations. Thus, if decoding is successful, it provides evidence for different patterns of spatial representation. That the decoding is successful was, to us and many others, surprising because MEG is not thought to have especially good spatial resolution. The success of the decoders therefor raises the possibility that MEG is sensitive to spatial representations that are much finer than previously recognized, which is a similar argument to the one made by Mark Stokes, that we cite. The upshot is that the prior conclusions about the spatial resolution of MEG seem to confound two issues: first, what is the finest difference in spatial pattern that MEG can pick up? Second, what is the resolution of spatial map that can be projected back on the cortex given an MEG signal? The answer to the first question could be that the difference in spatial pattern to which MEG is sensitive is very fine, even though the answer to the second question is that the resolution of the spatial map that can be projected on to the cortex using MEG measurements is relatively coarse.

The conclusion that “results provide partial support for the hypothesis” of daylight-locus is overstated. None of the results that evaluate this hypothesis were significant. Poorer decoding for blue is not consistent with this prediction, as it should have been both blue and orange. There are more parsimonious ways to explain blue being the odd colour, and the authors cite some of this literature. Thus, this is not partial support but rather means that this information is unlikely to be carried in the EEG signal, at least.

We believe that the results do provide support for the hypothesis, as argued in the response to the first comment above. We disagree that “none of the results that evaluate this hypothesis were significant.” To recap: The present manuscript explores the extent to which luminance contrast could be decoded from representations elicited by intermediate directions in color space—in this stimulus design, all stimuli modulate both S and L-M dimensions. If luminance representations are determined by the contribution of the S-cone mechanism, then luminance-polarity representations using stimuli that modulate the S cone with the same sign, regardless of the L-M contribution, should be comparably decodable. Contrary to this prediction, the results show a striking difference in luminance decoding when the stimuli modulate +S+(M-L) versus +S+(L-M). This asymmetry is not consistent with a simple interpretation of the contribution of the S mechanism to luminance representations, but it is consistent with the idea that luminance representations are asymmetric with respect to the daylight locus, even if a similar asymmetry between -S+(M-L) versus -S+(L-M) was not observed. In other words, the results show that only one pole of the daylight axis is compromised in encoding luminance contrast, the pole associated with the +S+(M-L) axis.

Reviewer #4 (Remarks to the Author):

I have read the revised documents from Hermann et al. and I have a few minor points that I think should be addressed:

(1) The authors subsample trials to equalise the peaks and make the signal onsets more comparable. While this is good practice, it still deserves a cautionary statement in the interpretation, as it only 'corrects' for the peak strength, and not other factors that could cause differences in decoding onsets, for example between trial temporal variability. Furthermore, the authors repeatedly attribute this procedure to Cichy et al., 2014, but this procedure was to the best of my knowledge not used or described in Cichy et al 2014. Perhaps the authors meant to cite Isik et al., 2014 instead who I believe were among the first to use this procedure in MEG decoding: Isik L, Meyers EM, Leibo JZ, Poggio T. The dynamics of invariant object recognition in the human visual system. *J Neurophysiol* 111, 91-102 (2014).

Thank you for your constructive feedback. The citation was, as you point out, incorrect: we intended to credit Isik with this approach, and have revised the manuscript accordingly. We have also added a note to recognize that the use of subsampling only accounts for differences in the magnitude of peak decoding on the onset of decoding accuracy, but other factors that could cause differences in decoding onsets, such as trial temporal variability, may not be accounted for. But importantly, the main conclusions about differences in timing of the representations of hue and luminance-contrast polarity are based on the time-to-peak decoding, not on the onset of significance of decoding, and the time-to-peak decoding did not change following subsampling.

(2) Several figures contain only 2 ticks on the axes (e.g. xaxis with 0ms and 600ms, and a yaxis of 50% and 85%). Increasing the number of ticks on those axes would make the results easier to interpret.

We have double checked and all the figures with graphs show multiple ticks, although not all tick marks are labeled with numbers. We had removed the numbers to make the figures look less cluttered, and justified the decision because the numbers for each tick are easily recovered given the number of tick marks and the limits. But we are happy to annotate the axes labels with more numbers if this is what is desired. We will defer to the editors.

(3) The authors need to carefully check the reference list. It contains several references to preprints that should be replaced with their published journal versions. I know that the preprints listed below have now been published, so they at least need updating, but I encourage the authors to do another pass before this paper is published. The list also contains a reference to the biorxiv version of the current manuscript.

- 88 Hermann K, Rosenthal I, Singh S, Pantazis D, Conway BR. Temporal dynamics of the neural mechanisms for encoding hue and luminance contrast uncovered by magnetoencephalography. *BioRxiv* <https://www.biorxiv.org/content/10.1101/2020.06.17.155713v2>, (2020).

o this seems to be the current manuscript

- 63 Grootswagers TR, A. K.; Shatek, S. M.; Carlson, T. A. . The neural dynamics underlying

prioritisation of task- relevant information. arXiv:210201303v2 (2021).

o published in NBDT

• 80 Hajonides JE, Nobre AC, van Ede F, Stokes MG. Decoding visual colour from scalp electroencephalography measurements. BioRxiv <https://doi.org/10.1101/2020.07.30.228437> (2020)

o published in neuroimage

• 107 King J-R, Wyart V. The Human Brain encodes a Chronicle of Visual Events at each Instant of Time. bioRxiv, 846576 (2019).

o published in journal of neuroscience

We have updated the references to reflect the publication of these preprints, and checked to see if there are other preprints that are now published that we cite. We have cited the biorxiv preprint of the present manuscript. Is this not appropriate? We will defer to the editors.

REVIEWER COMMENTS

Reviewer #3 (Remarks to the Author):

The authors insist that how luminance and colour contrast are processed when presented together “has not been definitively addressed in the literature on visual evoked potentials” and have presented a wealth of literature to them in my previous review. They now cover this literature and any further discussion of this topic is beyond this review. Nevertheless, I would invite the authors one final time to tone down some of their interpretations that are overstated and make it seem like these questions could not be addressed with VEPs and that they are somehow now been looked at for the first time thanks to information decoding.

In fact, by dividing and averaging single trials into several folds, decoding from EEG/MEG is essentially processing that same signal that is present in a VEP, but this signal is augmented by the fact that rather than collapsed into a univariate mean signal there is also the distribution of multiple EEG responses for the algorithm to learn from, placed within a multidimensional space in which all channels act as features. I am not sure if the authors have fully considered these links, as they claim that a latency modulation would not cause a significant change to decoding if it didn't also affect the spatial distribution of the signal in their response to my comments. To demonstrate to them that this would indeed be the case, I simulated two signals that differ only in the latency of the P300a signal, with one being 300 ms and the other 280ms, both projected from its standard source, which has a fixed location and orientation (i.e. it results in the same topography, just shifted in time). In the attached file, I present the ERP at electrode Fz and the outcomes of decoding across 64 head channels, where it can be seen that the latency difference produces a decodable alteration in the signals (essentially, the amplitude is shifted 20 ms forward which creates a measurable difference) despite the fact that the modulation of latency is univariate. Therefore, I insist that they must change the section of their manuscript which erroneously speculates about the nature of the spatial pattern of signals, as the temporal pattern is also relevant and cannot be neglected.

Coming back to the daylight locus interpretation, I remain unconvinced by the author's response. As saturation (at least in CIE Lab space) equals C/L, saturation is not even for the dark and light stimuli in this study; DKL space is furthermore very uneven in terms of saturation across hues (S-cone dimension is particularly impacted; see Schiller et al., 2017, Vis Res). Uneven integration of S-cone signals and luminance therefore cannot be discounted, and may be modulated by the dimension of saturation, which pulls on both colour and luminance contrast and is known to be adversely affected for the S-cone increment but not decrement (see discussion of that in Wool et al., 2015, JoV). Thus the authors' response incorrectly attests that there are no differences between S-(L+M) stimuli across the positive and negative polarity that can be explained by more simple, perceptual processes, rather than the author's preferred interpretation of alignment with daylight locus, which I still find difficult to accept as measurable with EEG to such a high degree, when EEG signals are known to rather be more attuned to pick up effects of contrast, or saturation.

The authors have also misinterpreted my comment to them about the parvocellular neurons processing luminance, rather than the magnocellular neurons, but this is probably my own fault, as I should have mentioned that these comments were based on the work of Reid and Shapley, which show that M pathway involvement is dominant at lower contrasts and lower frequencies but that for other stimuli P pathway contributes a lot too. The authors respond as if the main gist of my argument was that it is accepted knowledge that luminance and colour are encoded jointly – and yes, there are cortical units that encode both types of contrast together, such as colour-luminance neurons, as well as separate units for each - but my comment was getting at the fact that the luminance contrast of high-contrast, spatially modulated stimuli used in this study is much less likely to be encoded solely by magnocellular processes. This is the sense in which I disagree with the claim that M-pathway faster responding is responsible for faster luminance responses. It is common knowledge in VEP literature that luminance contrast speeds up responses by up to 20 ms, and this is the sense in which I argued that the authors' claims were not particularly novel in this regard.

ERP at Fz – the shift in latency by 20ms (black line, 280 ms peak) clearly causes a perturbation of amplitudes over the entire P300 window, leading to higher¹² amplitudes at the start, and lower later on)

This shift in amplitude is clearly decoable, see below (decoding from 64 channels)

November 23, 2021

Please find below our point-by-point responses in green, to reviewer comments.

REVIEWER COMMENTS

Reviewer #3 (Remarks to the Author):

The authors insist that how luminance and colour contrast are processed when presented together “has not been definitively addressed in the literature on visual evoked potentials” and have presented a wealth of literature to them in my previous review. They now cover this literature and any further discussion of this topic is beyond this review. Nevertheless, I would invite the authors one final time to tone down some of their interpretations that are overstated and make it seem like these questions could not be addressed with VEPs and that they are somehow now been looked at for the first time thanks to information decoding.

We are indebted to the reviewer for their constructive feedback, the collegiality of their reviews (despite disagreements in interpretation of the results), and the substantial time and thought that they have put into this process. We hope we have responded in kind, and that the paper is stronger for this process. We have tried once again to edit the manuscript to tone down any perceived over-reach in our claims, without jeopardizing the clarity of the motivation and arguments in the paper. These edits include revising the abstract to remove the assertion that the timing of hue and luminance representations are “not well established”, revising the introduction to remove the sentence “The time taken by the brain to encode hue and luminance contrast remains poorly understood”,

In fact, by dividing and averaging single trials into several folds, decoding from EEG/MEG is essentially processing that same signal that is present in a VEP, but this signal is augmented by the fact that rather than collapsed into a univariate mean signal there is also the distribution of multiple EEG responses for the algorithm to learn from, placed within a multidimensional space in which all channels act as features. I am not sure if the authors have fully considered these links, as they claim that a latency modulation would not cause a significant change to decoding if it didn't also affect the spatial distribution of the signal in their response to my comments. To demonstrate to them that this would indeed be the case, I simulated two signals that differ only in the latency of the P300a signal, with one being 300 ms and the other 280ms, both projected from its standard source, which has a fixed location and orientation (i.e. it results in the same topography, just shifted in time). In the attached file, I present the ERP at electrode Fz and the outcomes of decoding across 64 head channels, where it can be seen that the latency difference produces a decodable alteration in the signals (essentially, the amplitude is shifted 20 ms forward which creates a measurable difference) despite the fact that the modulation of latency is univariate. Therefore, I insist that they must change the section of their manuscript which erroneously speculates about the nature of the spatial pattern of signals, as the temporal pattern is also relevant and cannot be neglected.

We thank the reviewer for this thoughtful analysis. We think we now understand the point that the reviewer is making about the way in which latency differences could give rise to significant decoding. Two stimuli that elicit the exact same temporal response but with a different latency delay will be decodable from a univariate measure of each response (so long the temporal response has temporal structure) because at any given time point, there will be predictable differences in the magnitude of the responses. This is a useful point, and we have revised the manuscript to recognize this point. Of course, a simulation could also be created to show that differences in the spatial distribution of the responses would give rise to significant decoding.

We have revised the manuscript to explicitly state that stimuli that elicit different latency responses could be decoded from a univariate response. We have also revised the section in which we speculate about the nature of the spatial patterns, to recognize the tentative nature of this speculation.

Could the decoding results we present be accounted for simply by differences in the latency of responses for any (or all) of the various decoding problems described in the paper, for example through unintentional variation in the contrast of the stimuli (we recognize that contrast will impact the timing of the responses)? We recognize that our data do not provide a definitive answer, although we think this is nonetheless an important question for discussion (see ref. 26, 91).

The main reason why we wonder whether latency differences are insufficient to contribute to the decoding results is that each analysis (e.g. of the decoding of hue identity) involves averaging multiple separate decoding problems. For latency differences to be the explanation, one would have to invoke precisely the same latency differences in all the subproblems; and these latency differences would need to be comparable across sets of problems (e.g. decoding hue identity among dark colors and decoding hue identity among light colors), which seems unlikely.

Other reasons, which we have not included in the discussion, are that the latency differences would have to be immune to task since the peak decoding times were not impacted by task, which seems unlikely because, as others have shown, attentional mechanisms invoked by tasks like the one we used (Figure 4) would serve to change apparent contrast of the stimuli. Moreover, a simple difference in the temporal offset of two otherwise identical responses will probably bring about a fairly flat decoding function, an intuition that is supported by the analysis provided by the reviewer, which is not what the data show. At minimum, if latency differences are the explanation for the success of the decoders, any structure in the decoding function would likely not be meaningful since it would reflect the idiosyncratic temporal profiles of the responses. These predictions are at odds with the clear temporal structure evident in all the decoding curves in the paper.

We also note that in the subsequent comment, the reviewer is concerned about the possibility that the stimuli are “uneven in terms of saturation across hue”, which amounts to a concern that the stimuli vary in some metric of contrast (whatever metric this is, it cannot be cone contrast, because the stimuli were defined to have balanced cone contrast). If the stimuli do vary in contrast, and if the decoding is solely attributed to differences in latency of response, then averaging across decoding problems for different pairs of stimuli should be accompanied by different decoding time courses (because contrast will influence latency), which would not yield the consistent differences in decoding time course evident in the data.

The spatial scale of the representations for color and luminance polarity, and the extent to which decoding of them is feasible with MEG (or EEG), will have to await further study.

Coming back to the daylight locus interpretation, I remain unconvinced by the author's response. As saturation (at least in CIE Lab space) equals C/L, saturation is not even for the dark and light stimuli in this study; DKL space is furthermore very uneven in terms of saturation across hues (S-cone dimension is particularly impacted; see Schiller et al., 2017, Vis Res). Uneven integration of S-cone signals and luminance therefore cannot be discounted, and may be modulated by the dimension of saturation, which pulls on both colour and luminance contrast and is known to be adversely affected for the S-cone increment but not decrement (see discussion of that in Wool et al., 2015, JoV). Thus the authors' response incorrectly attests that there are no differences between S-(L+M) stimuli across the positive and negative polarity that can be explained by more simple, perceptual processes, rather than the author's preferred interpretation of alignment with daylight locus, which I still find difficult to accept as measurable with EEG to such a high degree, when EEG signals are known to rather be more attuned to pick up effects of contrast, or saturation.

We recognize that defining saturation is a ticklish business, and that it is likely that there are differences in the saturation of the stimuli at the two luminance polarity levels (with the darker stimuli having higher saturation); we acknowledged this point in the manuscript lines 253-257). We also recognize that there are asymmetries along the S cardinal axis that impact the S-cone increments differently from the S-cone decrements. Our stimulus design and analysis was done to avoid these potential challenges: First, we compared the decoding of luminance polarity for stimuli that were all S increments. This involved decoding luminance polarity from stimuli that were +S+(L-M) and +S+(M-L) (such stimuli appear bluish and pinkish; not bluish and yellowish as they would if we were comparing luminance decoding carried by S increments versus S decrements). Any difference in decoding luminance polarity carried by these two hues cannot be attributed to the well-documented asymmetry in luminance carried by +S versus -S, since both hues are S increments. (We also, compare decoding luminance polarity carried by the two S decrement hues, which similarly controls for any potential confound introduced by the well-known asymmetry across S increments versus S decrements). Second, any difference in saturation of the light versus the dark versions of the +S+(L-M) stimulus should be the same as any difference in saturation of the light versus the dark versions of the +S+(M-L) stimulus (the same goes for the S decrement hues). So, there is no reason to expect that saturation accounts for the difference in decoding luminance polarity using +S+(L-M) compared to +S+(M-L), since any possible saturation differences are the same for both sets of decoding problems. We have clarified this explanation in the discussion.

We were concerned that we have missed the point made by the reviewer and reached out to Qasim Zaidi, the senior author of the Wool study cited by the reviewer, and who had provided detailed comments on an earlier draft of the manuscript (as indicated in the acknowledgements). Qasim concurs with our reasoning. But one point is perhaps worth reiterating: we are not making an argument that the neural data are better explained by adaptation to the daylight locus rather than by perceptual processes. In fact, it is plausible that these explanations amount to the same thing, and we have revised the discussion to make this clarification. The key point here is that prior work has examined asymmetries defined in the cardinal cone axes, along the S axis (S increments versus S decrements). But the daylight locus is not aligned with this axis. The results in the paper show that the neural representations recovered by the decoding analysis reflect an asymmetry along the daylight locus not simply one along the cardinal S axis, and this asymmetry is in line with that documented in perception (e.g. see Winkler et al, Curr Biol 2015).

The authors have also misinterpreted my comment to them about the parvocellular neurons processing luminance, rather than the magnocellular neurons, but this is probably my own fault, as I should have mentioned that these comments were based on the work of Reid and Shapley, which show that M pathway involvement is dominant at lower contrasts and lower frequencies but that for other stimuli P pathway contributes a lot too. The authors respond as if the main gist of my argument was that it is accepted knowledge that luminance and colour are encoded jointly – and yes, there are cortical units that encode both types of contrast together, such as colour-luminance neurons, as well as separate units for each - but my comment was getting at the fact that the luminance contrast of high-contrast, spatially modulated stimuli used in this study is much less likely to be encoded solely by magnocellular processes. This is the sense in which I disagree with the claim that M-pathway faster responding is responsible for faster luminance responses. It is common knowledge in VEP literature that luminance contrast speeds up responses by up to 20 ms, and this is the sense in which I argued that the authors' claims were not particularly novel in this regard.

Thanks for this clarification. We agree that both M and P pathways are very likely responding to the stimuli we used. We suspect that part of the miscommunication on this point pertains to the difficulty in parsing out the extent to which a given LGN cell is responding to the “color” versus the “luminance”. We can sidestep this issue because the MEG signals are likely dominated by cortical activity, so the relevant hypotheses we can probe is the relative timing of luminance polarity and color representations in the cortex (and the extent to which these representations are invariant along the other dimension, hue or luminance polarity). We stand by the conclusion that the data support the hypothesis that the cortical representation of luminance is slightly earlier than the cortical representation of hue. Future work will have to settle the extent to which this observation holds across changes in contrast. At present, we suspect it does, since there was no difference in the timing of the representation of hue with changes in cone contrast (see Supplementary Figure 4): pairs of dark stimuli will have higher cone contrast in units of detection threshold compared to pairs of light stimuli, yet the decoding of hue among light stimuli showed the same time course as the decoding of dark stimuli. Moreover, the hue contrast of the stimuli was equal or arguably higher than the luminance-polarity contrast of the stimuli (and perceptually, the differences in hue were greater than the differences in luminance contrast as assessed behaviorally; see Supplementary Figure 5, which shows that detection in the 1-back-hue task was slightly more accurate or quicker than detection in the 1-back-luminance task). Thus a prediction based on contrast (or perceptual salience) would hold that the representation of hue would emerge earlier than the representation of luminance-polarity, which is the opposite of what the data show.

REVIEWER COMMENTS

Reviewer #3 (Remarks to the Author):

I am happy with the manuscript in its present form and would like to thank the authors for their earnest and patient engagement with my comments and suggestions. I also found the discussion collegiate and constructive, and am glad they had this impression too; I am excited about seeing further research on this topic from the authors' lab.

While I still disagree that systematic shifts in latency between conditions would not affect MEG/EEG decoding significantly, I think this section is now written in a manner that is more objective, and suggestive of various possibilities. Thus, it will stimulate research into this important and interesting question.